**A comprehensive study of hygroscopic properties of calcium- and magnesium-containing salts: implication for hygroscopicity of mineral dust and sea salt aerosols**

Liya Guo,[1,5,a] Wenjun Gu,[1,5,a] Chao Peng,[2,5] Weigang Wang,[2] Yong Jie Li,[3] Taomou Zong,[4] Yujing Tang,[1] Zhijun Wu,[4] Qinhao Lin,[1] Maofa Ge,[2,5,6] Guohua Zhang,[1] Min Hu,[4] Xinhui Bi,[1] Xinming Wang,[1,5,6] Mingjin Tang[1,5,6,*]

State Key Laboratory of Organic Geochemistry and Guangdong Key Laboratory of Environmental Protection and Resources Utilization, Guangzhou Institute of Geochemistry, Chinese Academy of Sciences, Guangzhou 510640, China
State Key Laboratory for Structural Chemistry of Unstable and Stable Species, Institute of Chemistry, Chinese Academy of Sciences, Beijing 100190, China
Department of Civil and Environmental Engineering, Faculty of Science and Technology, University of Macau, Avenida da Universidade, Taipa, Macau, China
State Key Joint Laboratory of Environmental Simulation and Pollution Control, College of Environmental Sciences and Engineering, Peking University, Beijing 100871, China
University of Chinese Academy of Sciences, Beijing 100049, China
Center for Excellence in Regional Atmospheric Environment, Institute of Urban Environment, Chinese Academy of Sciences, Xiamen 361021, China

[a] These two authors contributed equivalently to this work.

* Correspondence: Mingjin Tang (mingjintang@gig.ac.cn)

**Abstract**

Calcium- and magnesium-containing salts are important components for mineral dust and sea salt aerosols, but their physicochemical properties are not well understood yet. In this study, hygroscopic properties of eight Ca- and Mg-containing salts, including $Ca(NO_3)_2 \cdot 4H_2O$, $Mg(NO_3)_2 \cdot 6H_2O$, $MgCl_2 \cdot 6H_2O$, $CaCl_2 \cdot 6H_2O$, $Ca(HCOO)_2$, $Mg(HCOO)_2 \cdot 2H_2O$, $Ca(CH_3COO)_2 \cdot H_2O$ and $Mg(CH_3COO)_2 \cdot 4H_2O$, were investigated using two complementary techniques. A vapor sorption analyzer was used to measure the change of sample mass with relative humidity (RH) under isotherm conditions, and the deliquescence relative humidities (DRH) for temperature in the range of 5-30 $^{o}C$ as well as water-to-solute ratios as a function of RH at 5 and 25 $^{o}C$ were reported for these eight compounds. DRH values showed large variation for these compounds; for example, at 25 $^{o}C$ DRH were measured to be ~28.5% for $CaCl_2 \cdot 6H_2O$ and >95% for $Ca(HCOO)_2$ and $Mg(HCOO)_2 \cdot 2H_2O$. We further found that the dependence of DRH on temperature can be approximated by the Clausius-Clapeyron equation. In addition, a humidity-tandem differential mobility analyzer was used to measure the change in mobility diameter with RH (up to 90%) at room temperature, in order to determine hygroscopic growth factors of aerosol particles generated by atomizing water solutions of these eight compounds. All the aerosol particles studied in this work, very likely to be amorphous under dry conditions, started to grow at very low RH (as low as 10%) and showed continuous growth with RH. Hygroscopic growth factors at 90% RH were found to range from 1.26±0.04 for $Ca(HCOO)_2$ to 1.79±0.03 for $Ca(NO_3)_2$, and the single hygroscopicity parameter ranged from 0.09-0.13 for $Ca(CH_3COO)_2$ to 0.49-0.56 for $Ca(NO_3)_2$. Overall, our work provides a comprehensive investigation of hygroscopic properties of these Ca- and Mg-containing salts, largely improving our knowledge in the physicochemical properties of mineral dust and sea salt aerosols.

# 1 Introduction

Mineral dust, mainly emitted from arid and semi-arid regions with an annual flux of ~2000 Tg, is one of the most abundant types of aerosols in the troposphere (Textor et al., 2006; Ginoux et al., 2012). Mineral dust aerosol affects the climate system directly by scattering and absorbing solar and terrestrial radiation (Formenti et al., 2011; Ridley et al., 2016; Chen et al., 2017) and indirectly by serving as cloud condensation nuclei (CCN) and ice nucleating particles (INPs) (Hoose and Moehler, 2012; Creamean et al., 2013; Cziczo et al., 2013; Tang et al., 2016a). In addition, deposition of mineral dust particles is an important source of several nutrient elements (Fe and P, for example) for many ecosystems around the globe, thus having significant impacts on biogeochemical cycles in these regions (Jickells et al., 2005; Mahowald et al., 2009; Mahowald et al., 2011; Zhang et al., 2015).

Mineral dust aerosol has an average lifetime of 2-7 days in the atmosphere and can thus be transported over thousands of kilometers (Textor et al., 2006; Uno et al., 2009). During transport mineral dust particles may undergo heterogeneous reactions with trace gases, impacting the abundance of a number of important reactive trace gases both directly and indirectly (Usher et al., 2003; Crowley et al., 2010; Romanias et al., 2012; Tang et al., 2017). These reactions can also lead to change in chemical composition of mineral dust particles (Usher et al., 2003; Li and Shao, 2009; Li et al., 2010; Tang et al., 2012; Romanias et al., 2016) and thereby modification of their physicochemical and optical properties (Krueger et al., 2003; Vlasenko et al., 2006; Liu et al., 2008b; Sullivan et al., 2009; Tang et al., 2016a; Pan et al., 2017). Mineral dust particles contain substantial amounts of carbonates, including $CaCO_3$ (calcite) and $CaMg(CO_3)_2$ (dolomite) (Nickovic et al., 2012; Formenti et al., 2014; Jeong and Achterberg, 2014; Journet et al., 2014; Scanza et al., 2015). These carbonates are largely insoluble and have very low hygroscopicity

(Sullivan et al., 2009; Tang et al., 2016a); however, their reactions with acidic gases in the
troposphere can form Ca- and Mg-containing salts with higher hygroscopicity (Gibson et al., 2006;
Liu et al., 2008b; Sullivan et al., 2009; Tang et al., 2016a), such as $Ca(NO_3)_2$ and $Mg(NO_3)_2$. For
example, numerous laboratory and field studies have found that due to the formation of $Ca(NO_3)_2$
and $CaCl_2$ from heterogeneous reactions with nitrogen oxides (Goodman et al., 2000; Liu et al.,
2008a; Li et al., 2010; Tang et al., 2012; Tan et al., 2016) and HCl (Santschi and Rossi, 2006),
solid $CaCO_3$ particles could be converted to aqueous droplets under tropospheric conditions
(Krueger et al., 2003; Laskin et al., 2005; Liu et al., 2008b; Shi et al., 2008; Tobo et al., 2010). In
addition, $MgCl_2$ and $CaCl_2$ are important components in sea salt aerosol (as known as sea spray
aerosol). The presence of $MgCl_2$ and $CaCl_2$, in addition to NaCl, can alter the hygroscopicity of
sea salt aerosol (Gupta et al., 2015; Zieger et al., 2017); to be more specific, the hygroscopicity of
sea salt was found to be significantly smaller than pure NaCl. Furthermore, the CCN activity of
saline mineral dust was explored (Gaston et al., 2017), and good correlations were found between
the CCN activities of saline mineral dust particles and the abundance of the soluble components
(e.g., $CaCl_2$) they contained.
Nevertheless, hygroscopic properties of $Ca(NO_3)_2$, $Mg(NO_3)_2$, $CaCl_2$ and $MgCl_2$ have not
been completely understood, especially in the two following aspects. First, hygroscopic growth
factors were only measured by one or two previous studies for $Ca(NO_3)_2$ (Gibson et al., 2006; Jing
et al., 2018), $Mg(NO_3)_2$ (Gibson et al., 2006), $CaCl_2$ (Park et al., 2009) and $MgCl_2$ aerosols (Park
et al., 2009). Considering the importance of these compounds in the troposphere, additional
measurements of their hygroscopic growth are clearly warranted. In addition, tropospheric
temperatures range from ~200 to ~300 K; however, the effects of temperature on their phase

transitions and hygroscopic growth remain largely unclear (Kelly and Wexler, 2005), due to lack of experimental data below room temperature.

Small carboxylic acids, such as formic and acetic acids, are abundant in the troposphere (Khare et al., 1999), and previous studies suggested that heterogeneous reactions of mineral dust with formic and acetic acids are efficient (Hatch et al., 2007; Prince et al., 2008; Tong et al., 2010; Ma et al., 2012; Tang et al., 2016b). It was shown that calcium and magnesium acetates were formed in heterogeneous reactions of gaseous acetic acid with MgO and $CaCO_3$ particles, leading to significant increase in particle hygroscopicity (Ma et al., 2012). However, only a few previous studies explored hygroscopic growth of $Mg(CH_3COO)_2$ and $Ca(CH_3COO)_2$, using techniques based on bulk samples (Wang et al., 2005; Ma et al., 2012; Pang et al., 2015). To our knowledge, hygroscopic growth factors have never been reported for $Ca(HCOO)_2$, $Mg(HCOO)_2$, $Ca(CH_3COO)_2$ and $Mg(CH_3COO)_2$ aerosol particles.

To better understand hygroscopic properties of these Ca- and Mg-containing salts, two complementary techniques were employed in this work to investigate their phase transitions and hygroscopic growth. A vapor sorption analyzer, which measured the sample mass as a function of RH, was used to determine the DRH and solute-to-water ratios for $Ca(NO_3)_2 \cdot 4H_2O$, $Mg(NO_3)_2 \cdot 6H_2O$, $CaCl_2 \cdot 6H_2O$, $MgCl_2 \cdot 6H_2O$, $Ca(HCOO)_2$, $Mg(HCOO)_2 \cdot 2H_2O$, $Ca(CH_3COO)_2 \cdot H_2O$ and $Mg(CH_3COO)_2 \cdot 4H_2O$ at different temperatures (5-30 $^oC$). Furthermore, hygroscopic growth factors of $Ca(NO_3)_2$, $Mg(NO_3)_2$, $CaCl_2$, $MgCl_2$, $Ca(HCOO)_2$, $Mg(HCOO)_2$, $Ca(CH_3COO)_2$ and $Mg(CH_3COO)_2$ aerosol particles were determined at room temperature up to 90% RH, using a humidity-tandem differential mobility analyzer. This work would significantly increase our knowledge in the hygroscopicity of these compounds, hence leading to a better understanding of the physicochemical properties of mineral dust and sea salt aerosols.

## 2 Experimental section

Hygroscopic growth of Ca- and Mg-containing salts were investigated using two complementary techniques, i.e. a humidity-tandem differential mobility analyzer (H-TDMA) and a vapor sorption analyzer (VSA). Eight salts, all supplied by Aldrich, were investigated in this work, including $Ca(NO_3)_2 \cdot 4H_2O$ (>99%), $Mg(NO_3)_2 \cdot 6H_2O$ (99%), $CaCl_2 \cdot 6H_2O$ (>99%), $MgCl_2 \cdot 6H_2O$ (>99%), $Ca(HCOO)_2$ (>99%), $Mg(HCOO)_2 \cdot 2H_2O$ (98%), $Ca(CH_3COO)_2 \cdot H_2O$ (>99%) and $Mg(CH_3COO)_2 \cdot 4H_2O$ (99%).

### 2.1 H-TDMA experiments

H-TDMA measurements were carried out at Institute of Chemistry, Chinese Academy of Sciences, and the experimental setup was detailed in previous work (Lei et al., 2014; Peng et al., 2016). Hygroscopic growth of size-selected aerosol particles was determined by measuring their mobility diameters at different RH. An atomizer (MSP 1500) was used to generate aerosol particles. Solutions used for atomization were prepared using ultrapure water, and their typical concentrations were 0.3-0.4 g $L^{-1}$. After exiting the atomizer, an aerosol flow (300 mL/min) was passed through a Nafion dryer and then a diffusion dryer filled with silica gel to reach a final RH of <5%. The aerosol flow was then delivered through a neutralizer and the first differential mobility analyzer (DMA) to produce quasi-monodisperse aerosol particles with a mobility diameter of 100 nm. After that, the aerosol flow was transferred through a humidification section with a residence time of ~27 s to be humidified to a given RH. The humidification section was made of two Nafion humidifiers (MD-700-12F-1, Perma Pure) connected in series. The RH of the resulting aerosol flow was monitored using a dew-point meter, which had an absolute uncertainty of ±0.8% in RH measurement as stated by the manufacturer (Michell, UK). After humidification, the size distribution of aerosol particles was measured using a scanning mobility particle sizer

(SMPS) which consisted of the second DMA coupled with a condensation particle counter (TSI
3776). For the second DMA, the aerosol flow and the sheath flow were always maintained at the
same RH. The flow rate ratios of the aerosol flow to the sheath flow were set to 1:10 for both
DMA.
In our work, the hygroscopic growth factor (GF) is defined as the ratio of measured
mobility diameters at a given RH to that at dry conditions:
$$GF = \frac{d}{d_0} \qquad (1)$$

where $d_0$ and $d$ are the measured mobility diameters at <5% RH and at a given RH, respectively.
In our work the dry mobility diameter selected using the first DMA was always 100 nm, and no
shape factors were used to correct the dry particle diameters. Size distributions of all the eight
types of aerosol particles, measured using the SMPS, were found to be unimode, as illustrated by
Figure S1 (in the supplementary information) in which size distributions of $Ca(NO_3)_2$ aerosols at
4, 50 and 90% RH are displayed as an example. The TDMAinv algorithm (Gysel et al., 2009) was
applied to the H-TDMA data.
All the experiments were carried out at room temperature (298±1 K), and in each
experiment hygroscopic growth of aerosol particles was determined at 12 different RH, i.e. <5, 10,
20, 30, 40, 50, 60, 70, 75, 80, 85, and 90%. The absolute uncertainties in RH were estimated to be
within ±2%. Hygroscopic growth of each compound was measured three times. The performance
of the H-TDMA setup was routinely checked by measuring the hygroscopic growth of 100 nm
$(NH_4)_2SO_4$ and NaCl aerosol particles. Good agreement between measured hygroscopic growth
curves with those predicted using the E-AIM model (Clegg et al., 1998) was always found for
$(NH_4)_2SO_4$ and NaCl aerosols, as detailed in our previous work (Jing et al., 2016; Peng et al., 2016).

## 2.2 VSA experiments

The vapor sorption analyzer (Q5000SA), which measured the mass of a bulk sample as a function of RH under isotherm conditions, was manufactured by TA Instruments (New Castle, DE, USA). These experiments were performed at Guangzhou Institute of Geochemistry, Chinese Academy of Sciences, and the instrument and experimental method were described elsewhere (Gu et al., 2017a; Gu et al., 2017b; Jia et al., 2018). Experiments could be conducted in a temperature range of 5-85 $^{\circ}$C with an accuracy of $\pm0.1$ $^{\circ}$C and a RH range of 0-98% with an absolute accuracy of $\pm1$%. The mass measurement had a range of 0-100 mg, and its sensitivity was stated to be <0.1 μg. Initial mass of samples used in an experiment was usually in the range of 0.5-1 mg.

Two different types of experiments were carried out. The mass hygroscopic growth was studied in the first type of experiments: after the sample was dried at <1% RH as a given temperature, RH was increased to 90% stepwise with an increment of 10% per step; after that, RH was set to 0% (the actual RH was measured to be <1%) to dry the sample again. The second type of experiments were conducted to measure DRH values: the sample was first dried at a given temperature, and RH was increased to a value which was at least 5% lower than the expected DRH; RH was then increased stepwise with an increment of 1% until a significant increase in sample mass was observed, and the RH at which the sample mass showed a significant increase was equal to its DRH. The measured relative change in sample mass due to signal noise and baseline drift was <0.5% in our work; in each experiment when we suspected that the sample were undergoing deliquescence at a certain RH, we did not stop the experiment until the mass increase was >5% to ensure the occurrence of deliquescence. At each RH the sample was considered to reach equilibrium with the environment when its mass change was <0.1% within 30 min, and RH was changed to the next value only after the sample mass was stabilized. If the sample mass was

increasing steadily but with a very small rate (e.g., <0.1% in 30 min), the program we used may
conclude erroneously that the system had reached the equilibrium; therefore, all the experimental
data were inspected to check whether at each RH the sample mass reached the plateau (i.e. the
system had reached the equilibrium). The time to reach a new equilibrium varied with compounds
and largely depended on the dry sample mass, i.e. a sample with larger dry mass would took longer
to reach the equilibrium. Each experiment was repeated at least three times, and the average value
and standard deviation were reported.

## 3 Results and discussion

### 3.1 Hygroscopicity of nitrates and chlorides

**3.1.1 DRH at different temperature**

First we investigated the effect of temperature on the DRH of $Ca(NO_3)_2 \cdot 4H_2O$,
$Mg(NO_3)_2 \cdot 6H_2O$ and $MgCl_2 \cdot 6H_2O$, which are the most stable forms of corresponding salts for the
temperature range (5-30 $^o$C) considered in this work (Kelly and Wexler, 2005). Figure 1a shows
the change of RH and normalized sample mass as a function of time in an experiment to measure
the DRH of $Mg(NO_3)_2 \cdot 6H_2O$ at 25 $^o$C. Abrupt and significant increase in sample mass was
observed when RH was increased from 52 to 53%, suggesting that the deliquescence occurred
between 52 and 53% RH. Therefore, its DRH was measured to be 52.5±0.5 %; since RH for our
VSA instrument had an absolute uncertainty of ±1% (as stated in Section 2.2), in our work an
uncertainty of ±1%, instead of ±0.5%, was assigned to the measured DRH. It should be noted that
the mass change was >15% when RH was increased from 52 to 53%, as shown in Figure 1a; such
a large mass increase cannot be solely caused by water adsorption, since the mass of several
monolayers of adsorbed water is estimated to be <1% of the dry particle mass (Gu et al., 2017b).
The continuous but small decrease in sample mass (about 1% in total) with time (around 500-1000
min) before deliquescence took place, as shown in Figure 1a, was likely caused by desorption of
residual water contained by the sample under investigation.

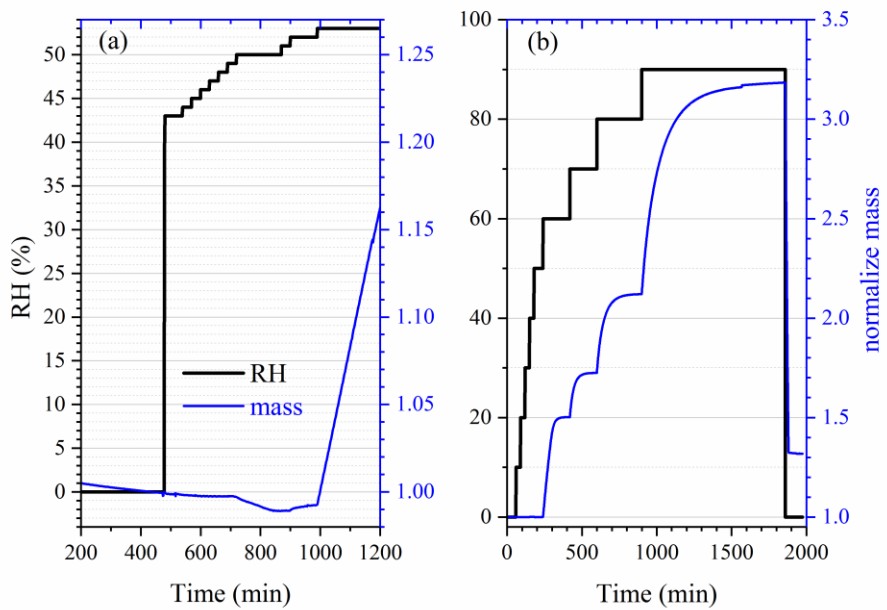


**Figure 1.** Change of normalized sample mass (blue curve, right *y*-axis) and RH (black curve, left
*y*-axis) as a function of time. (a) A typical experiment conducted to measure the DRH; (b) A typical
experiment conducted to measure mass hygroscopic growth factors. In the two experiments shown
here, $Mg(NO_3)_2 \cdot 6H_2O$ was investigated at 25 ℃. In this paper the sample mass was always
normalized to its dry mass.

Table 1 summarizes our measured DRH of $Ca(NO_3)_2 \cdot 4H_2O$, $Mg(NO_3)_2 \cdot 6H_2O$ and
$MgCl_2 \cdot 6H_2O$ as a function of temperature (5-30 ℃). DRH values show a strong dependence on
temperature for $Ca(NO_3)_2 \cdot 4H_2O$ (decreasing from 60.5% at 5 ℃ to 46.0% at 30 ℃) and a weaker
temperature dependence for $Mg(NO_3)_2 \cdot 6H_2O$ (decreasing from 57.5% at 5 ℃ to 50.5% at 30 ℃);
in contrast, the DRH values of $MgCl_2 \cdot 6H_2O$ (31.5-32.5 %) exhibit little variation with temperature
(5-30 °C). Several previous studies have reported the DRH of $Ca(NO_3)_2 \cdot 4H_2O$, $Mg(NO_3)_2 \cdot 6H_2O$
and $MgCl_2 \cdot 6H_2O$, and their results are compared with our work in the following paragraphs.

**Table 1.** DRH (in %) of $Ca(NO_3)_2 \cdot 4H_2O$, $Mg(NO_3)_2 \cdot 6H_2O$ and $MgCl_2 \cdot 6H_2O$ measured in this work
as a function of temperatures (5-30 °C). Solubility data (mol per kg water) compiled by Kelly and
Wexler (2005) was used to calculate solubilities in mol per mol water. All the errors given in this
work are standard deviations.

| $T$ (°C) | $Ca(NO_3)_2 \cdot 4H_2O$ | $Mg(NO_3)_2 \cdot 6H_2O$ | $MgCl_2 \cdot 6H_2O$ |
|---|---|---|---|
| 5 | 60.5±1.0 | 57.5±1.0 | 32.5±1.0 |
| 10 | 58.0±1.0 | 56.5±1.0 | 32.5±1.0 |
| 15 | 55.5±1.0 | 54.5±1.0 | 32.5±1.0 |
| 20 | 52.5±1.0 | 53.5±1.0 | 32.5±1.0 |
| 25 | 49.5±1.0 | 52.5±1.0 | 31.5±1.0 |
| 30 | 46.0±1.0 | 50.5±1.0 | 31.5±1.0 |
| solubility (mol per kg water) | 8.4 | 4.9 | 5.84 |
| solubility ($A$, mol per mol water) | 0.1512 | 0.0882 | 0.1051 |
| $A \cdot \Delta H_s/R$ (K) | 913±59 | 427±28 | -- |
| $\Delta H_s$ (kJ mol$^{-1}$) | 50.2±3.3 | 40.3±2.6 | -- |

The $A \cdot \Delta H_s/R$ and $\Delta H_s$ values were not estimated for $MgCl_2 \cdot 6H_2O$ because the difference in its measured
DRH between 5 and 30 °C was very small or even insignificant. Please refer to Section 3.1.1 for further
details.

**$Ca(NO_3)_2 \cdot 4H_2O$:** RH of air in equilibrium with saturated $Ca(NO_3)_2 \cdot 4H_2O$ solutions, i.e.

the DRH values of $Ca(NO_3)_2 \cdot 4H_2O$, were measured to be 55.9, 55.4, 50.5 and 46.7% at 15, 20, 25
and 30 °C (Adams and Merz, 1929), and the absolute differences between DRH reported by Adams
and Merz (1929) and those measured in our work are <3%. The water vapor pressures of saturated
$Ca(NO_3)_2 \cdot 4H_2O$ solutions were measured to be 0.693, 0.920, 1.253, 1.591 and 1.986 kPa at 10, 15,
20, 25 and 30 °C (Apelblat, 1992), corresponding to DRH of 56, 54, 54, 50 and 47%, respectively;
therefore, the absolute difference between DRH measured in our work and those derived from
Apelblat (1992) are <2%. In another study (Al-Abadleh et al., 2003), RH over the saturated
$Ca(NO_3)_2 \cdot 4H_2O$ solution was measured to be 57±5% at room temperature; in other words, Al-
Abadleh et al. (2003) reported a DRH of 57±5% for $Ca(NO_3)_2 \cdot 4H_2O$, slightly larger than that
(49.5±1.0% at 25 °C) determined in our work.
**$Mg(NO_3)_2 \cdot 6H_2O$:** Water vapor pressures of saturated $Mg(NO_3)_2 \cdot 6H_2O$ solutions were
determined to be 0.737, 1.017, 1.390, 1.813 and 2.306 kPa at 10, 15, 20, 25 and 30 °C (Apelblat,
1992), giving DRH of 60, 60, 59, 57 and 54% at corresponding temperatures. The vapor pressure
of saturated $Mg(NO_3)_2 \cdot 6H_2O$ solutions at 25 °C were reported to be 1.674 and 1.666 kPa by another
two studies (Biggs et al., 1955; Robinson and Stokes, 1959), corresponding to DRH of ~53%. In
addition, the water activity of the saturated $Mg(NO_3)_2$ solution was measured to be 0.528 at 25 °C
(Rard et al., 2004), also suggesting a DRH value of ~53%; similarly, RH over the saturated
$Mg(NO_3)_2$ solution was reported to be ~53% at 22-24 °C (Li et al., 2008b). Al-Abadleh and
Grassian (2003) investigated the phase transition of the $Mg(NO_3)_2 \cdot 6H_2O$ film, and its DRH was
determined to be 49-54% at 23 °C. As shown in Table 1, DRH measured in our work agree very
well with those reported by most of previous studies (Biggs et al., 1955; Robinson and Stokes,
1959; Al-Abadleh and Grassian, 2003; Rard et al., 2004), but are always 3-5% lower than those
derived from Apelblat (1992). It is not clear why DRH values measured by Apelblat (1992) at
different temperatures are always slightly higher than other studies.
**$MgCl_2 \cdot 6H_2O$:** Kelly and Wexler (2005) calculated DRH of $MgCl_2 \cdot 6H_2O$ from vapor
pressures of saturated $MgCl_2 \cdot 6H_2O$ solutions measured by previous work, and found that DRH
values were in the range of 33-34% for temperatures at 0-40 °C. In addition, water activity of the
saturated MgCl$_2$ solution was reported to be 0.3278 at 25 $^o$C (Rard and Miller, 1981),
corresponding to a DRH value of ~33% for MgCl$_2$·6H$_2$O. The DRH values of MgCl$_2$·6H$_2$O
measured in our work, as summarized in Table 1, show excellent agreement with those reported
by previous work (Rard and Miller, 1981; Kelly and Wexler, 2005). Phase transition and
deliquescence behavior of CaCl$_2$·6H$_2$O were also investigated in our work and found to be very
complex, and the result will be discussed in Section 3.1.3.

Temperature in the troposphere varies from ~200 to >300 K, and it is thus warranted to

explore the effects of temperature on hygroscopic properties of atmospherically relevant particles.
The dependence of DRH on temperature can usually be approximated by the Clausius-Clapeyron
equation (Wexler and Seinfeld, 1991; Seinfeld and Pandis, 2016; Jia et al., 2018):
$$\ln[DRH(T)] = \ln[\text{DRH}(298)] + \frac{A \cdot \Delta H_S}{R}(\frac{1}{T} - \frac{1}{298}) \quad (2)$$
where $T$ is temperature (K), DRH($T$) and DRH(298) are the DRH at $T$ and 298 K, $R$ is the gas
constant (8.314 J mol$^{-1}$ K$^{-1}$), and $\Delta H_s$ is the enthalpy of dissolution (J mol$^{-1}$). The dimensionless
constant, $A$, is numerically equal to the water solubility of the salt under investigation in the unit
of mol per mol water. Figure 2 shows the dependence of DRH values on temperature for
Ca(NO$_3$)$_2$·4H$_2$O and Mg(NO$_3$)$_2$·6H$_2$O, confirming that Eq. (2) can indeed approximate the
temperature dependence. The slope, which is equal to $A$·$\Delta H_s$/$R$, was determined to be 913±59 K
for Ca(NO$_3$)$_2$·4H$_2$O and 427±28 K for Mg(NO$_3$)$_2$·6H$_2$O, and thus $\Delta H_s$ was derived to be 50.2±3.3
kJ mol$^{-1}$ for Ca(NO$_3$)$_2$·4H$_2$O and 40.3±2.6 kJ mol$^{-1}$ for Mg(NO$_3$)$_2$·6H$_2$O. It should be noted that
for Eq. (2) to be valid, both the enthalpy of dissolution and the water solubility are assumed to be
constant for the temperature range considered. The variation of DRH with temperature (5-30 $^o$C)
was very small and even insignificant for MgCl$_2$·6H$_2$O; as a result, we did not attempt to estimate
the $\Delta H_s$ values for MgCl$_2$·6H$_2$O since such estimation would have large errors.

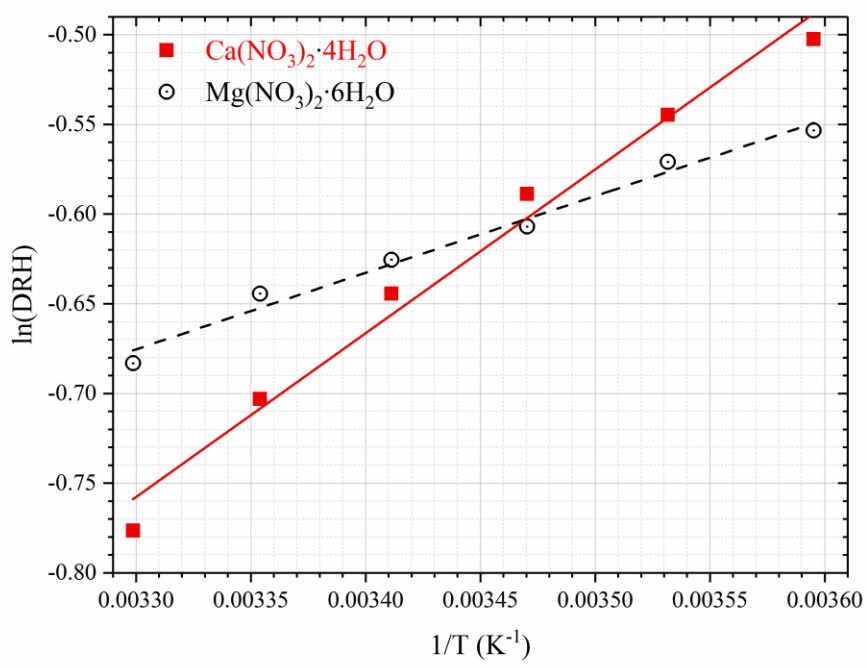


**Figure 2.** Dependence of DRH on temperature for $Ca(NO_3)_2 \cdot 4H_2O$ and $Mg(NO_3)_2 \cdot 6H_2O$.

### 3.1.2 Water-to-solute ratios as a function of RH

The change of sample mass with RH (0-90%) was measured at 5 and 25 $^{\circ}$C for $Ca(NO_3)_2 \cdot 4H_2O$, $Mg(NO_3)_2 \cdot 6H_2O$ and $MgCl_2 \cdot 6H_2O$, using the vapor sorption analyzer. The mass change, relative to that at 0% RH, can be used to calculate water-to-solute ratios (WSR, defined in this work as the molar ratio of $H_2O$ to $Ca^{2+}$ or $Mg^{2+}$) for deliquesced samples. Small increases in $m/m_0$ (typically <2%) were observed for some compounds (as shown in Tables 2 and 6) when RH was below corresponding DRH values, mainly due to water adsorption/desorption and baseline drift. As summarized in Table 2, decrease in temperature would lead to increase in WSR at a given RH: at 90% RH for example, WSR were determined to be 28.78±0.20 at 25 $^{\circ}$C and 31.80±0.96 at 5 $^{\circ}$C for $Ca(NO_3)_2 \cdot 4H_2O$, 36.87±0.23 at 25 $^{\circ}$C and 41.40±1.36 at 5 $^{\circ}$C for $Mg(NO_3)_2 \cdot 6H_2O$, and 36.26±1.76 at 25 $^{\circ}$C and 39.55±2.43 at 5 $^{\circ}$C for $MgCl_2 \cdot 6H_2O$, respectively. As discussed in Section 3.1.1, the enthalpies of dissolution ($\Delta H_s$) are negative for these compounds, suggesting that their

dissolution processes in water are exothermic; therefore, dissolution is favored at lower
temperatures and at a given RH, decrease in temperature would lead to increase in WSR in the
aqueous solutions. Several previous studies have measured RH over aqueous $Ca(NO_3)_2$, $Mg(NO_3)_2$
and $MgCl_2$ solutions at given concentrations, and their results are compared with our work, as
discussed below.

**Table 2.** Mass growth factors ($m/m_0$, defined as the ratio of sample mass at a given RH to that at
0% RH) and water-to-solute ratios (WSR) as a function of RH (0-90%) at 25 and 5 °C for
$Ca(NO_3)_2 \cdot 4H_2O$, $Mg(NO_3)_2 \cdot 6H_2O$ and $MgCl_2 \cdot 6H_2O$. WSR were only calculated for RH exceeding
the DRH (i.e. when the sample was deliquesced). All the errors given in this work are standard
deviations.

| | $Ca(NO_3)_2 \cdot 4H_2O$, 25 °C | | $Ca(NO_3)_2 \cdot 4H_2O$, 5 °C | |
|---|---|---|---|---|
| RH (%) | $m/m_0$ | WSR | $m/m_0$ | WSR |
| 0 | 1.000±0.001 | -- | 1.000±0.001 | -- |
| 10 | 1.000±0.001 | -- | 1.001±0.001 | -- |
| 20 | 1.014±0.005 | -- | 1.005±0.003 | -- |
| 30 | 1.016±0.007 | -- | 1.005±0.002 | -- |
| 40 | 1.017±0.009 | -- | 1.009±0.003 | -- |
| 50 | 1.237±0.006 | 7.10±0.03 | 1.032±0.005 | -- |
| 60 | 1.363±0.008 | 8.76±0.05 | 1.041±0.002 | -- |
| 70 | 1.550±0.009 | 11.22±0.06 | 1.610±0.010 | 12.00±0.07 |
| 80 | 1.897±0.012 | 15.77±0.10 | 1.979±0.027 | 16.85±0.23 |
| 90 | 2.889±0.020 | 28.78±0.20 | 3.119±0.095 | 31.80±0.96 |
| | $Mg(NO_3)_2 \cdot 6H_2O$, 25 °C | | $Mg(NO_3)_2 \cdot 6H_2O$, 5 °C | |
| RH (%) | $m/m_0$ | WSR | $m/m_0$ | WSR |
| 0 | 1.000±0.001 | -- | 1.000±0.001 | -- |
| 10 | 1.000±0.001 | -- | 1.000±0.001 | -- |
| 20 | 1.000±0.001 | -- | 1.000±0.001 | -- |

| | | | | |
|---|---|---|---|---|
| 30 | 1.001±0.001 | -- | 1.000±0.001 | -- |
| 40 | 1.001±0.001 | -- | 1.000±0.001 | -- |
| 50 | 1.000±0.001 | -- | 1.000±0.001 | -- |
| 60 | 1.503±0.001 | 13.15±0.01 | 1.539±0.003 | 13.67±0.03 |
| 70 | 1.724±0.001 | 16.30±0.01 | 1.773±0.007 | 16.99±0.07 |
| 80 | 2.121±0.001 | 21.94±0.01 | 2.203±0.021 | 23.11±0.22 |
| 90 | 3.171±0.029 | 36.87±0.23 | 3.489±0.114 | 41.40±1.36 |

| $MgCl_2 \cdot 6H_2O$, 25 °C | | $MgCl_2 \cdot 6H_2O$, 5 °C | |
|---|---|---|---|
| RH (%) | $m/m_0$ | WSR | $m/m_0$ | WSR |
|---|---|---|---|---|
| 0 | 1.000±0.001 | -- | 1.000±0.001 | -- |
| 10 | 1.000±0.001 | -- | 1.000±0.001 | -- |
| 20 | 1.000±0.001 | -- | 1.000±0.001 | -- |
| 30 | 1.001±0.001 | -- | 1.000±0.001 | -- |
| 40 | 1.344±0.057 | 9.89±0.42 | 1.327±0.082 | 9.69±0.60 |
| 50 | 1.489±0.062 | 11.52±0.48 | 1.473±0.090 | 11.34±0.69 |
| 60 | 1.677±0.072 | 13.65±0.58 | 1.667±0.100 | 13.52±0.82 |
| 70 | 1.951±0.084 | 16.74±0.72 | 1.950±0.117 | 16.72±1.00 |
| 80 | 2.433±0.117 | 22.18±1.06 | 2.465±0.148 | 22.54±1.35 |
| 90 | 3.681±0.178 | 36.26±1.76 | 3.972±0.244 | 39.55±2.43 |

**Ca(NO₃)₂:** Water activities of $Ca(NO_3)_2$ solutions at 25 °C were measured to be 0.904, 0.812 and 0.712 when the concentrations were 2.0, 3.5 and 5.0 mol kg[-1], respectively (El Guendouzi and Marouani, 2003). Since water activity of a solution is equal to the RH of air in equilibrium with the solution, it can be derived that the molality concentrations of $Ca(NO_3)_2$ solution were 2.0, 3.5 and 5.0 mol kg[-1] when RH was 90.4, 81.2 and 71.2%; in other words, WSR were found to be 11.1, 15.9 and 27.8 at 71.2, 81.2 and 90.4 % RH, respectively (El Guendouzi and Marouani, 2003). As shown in Table 2, in our work WSR were determined to be 11.22±0.06, 15.77±0.10 and 28.78±0.20 at 70, 80 and 90% RH for $Ca(NO_3)_2$ solutions at the same temperature, suggesting good agreement with El Guendouzi and Marouani (2003).

**Mg(NO₃)₂:** Water activities of $Mg(NO_3)_2$ solutions were reported to be 0.897, 0.812 and
0.702 when the concentrations of the bulk solutions were 1.6, 2.5 and 3.5 mol kg$^{-1}$ at 25 $^{\circ}$C,
respectively (Rard et al., 2004); this means that WSR were equal to 15.9, 22.2 and 34.7 at 70.2,
81.2 and 89.7% RH. Ha and Chan (1999) fitted their measured water activities of $Mg(NO_3)_2$ as a
function of molality concentration at 20-24 $^{\circ}$C with a polynomial equation, and WSR were derived
to be 12.93, 16.12, 21.50 and 36.09 at 60, 70, 80 and 90% RH. As shown in Table 2, WSR were
measured to be 13.15±0.01, 16.30±0.01, 21.94±0.01 and 36.87±0.23 at 60, 70, 80 and 90% RH
for deliquesced $Mg(NO_3)_2$ at 25 $^{\circ}$C. Therefore, it can be concluded that for WSR of $Mg(NO_3)_2$
solutions at ~25 $^{\circ}$C, our work shows good agreement with the two previous studies (Ha and Chan,
1999; Rard et al., 2004).
**MgCl₂:** Water activities of $MgCl_2$ solutions were reported to be 0.909, 0.800, 0.692, 0.491
and 0.408 when the concentrations were 1.4, 2.4, 3.2, 4.6 and 5.2 mol kg$^{-1}$ (Rard and Miller, 1981),
i.e. WSR were equal to 10.7, 12.1, 17.4, 23.1 and 39.7 at 40.8, 49.1, 69.2, 80.0 and 90.9% RH. In
another work  (Ha and Chan, 1999), an electrodynamic balance was used to investigate
hygroscopic growth of $MgCl_2$ particles at 20-24 $^{\circ}$C, and the measured molality concentrations of
$MgCl_2$ solutions as a function of water activity were fitted by a polynomial equation. It can be
derived from Ha and Chen (1999) that WSR were equal to 10.65, 12.34, 14.29, 17.04, 22.24 and
34.78 when RH were 40, 50, 60, 70, 80 and 90%, respectively. WSR measured in our work, as
listed in Table 2, are 9.89±0.42, 11.52±0.48, 1.677±0.072, 16.74±0.72, 22.18±1.06 and 36.26±1.76
at 40, 50, 60, 70, 80 and 90% RH. As a result, our work agrees well with the two previous studies
(Rard and Miller, 1981; Ha and Chan, 1999) for WSR of $MgCl_2$ solutions as a function of RH at
~25 $^{\circ}$C.
**3.1.3 Phase transition of CaCl₂·$x$H₂O**

The change in sample mass of $CaCl_2·6H_2O$ with RH was also investigated at 25 °C. As shown in Figure 3, when dried at 0% RH, the sample mass was reduced by 1/3 (from ~1.5 to ~1.0), and it is speculated that $CaCl_2·6H_2O$ was converted to $CaCl_2·2H_2O$. When RH was increased to 10%, no significant increase in sample mass was observed. As RH was further increased to 20%, the sample mass was increased by 48±7 %; this may indicate that $CaCl_2·2H_2O$ was converted to $CaCl_2·6H_2O$, as the ratio of molar mass of $CaCl_2·6H_2O$ (219 g mol$^{-1}$) to $CaCl_2·2H_2O$ (147 g mol$^{-1}$) is 1.49, approximately equal to the ratio of sample mass at 20% RH to that at 10% RH. Further increase in RH to 30% would lead to additional increase in sample mass, implying the deliquescence of the sample and the formation of an aqueous $CaCl_2$ solution.

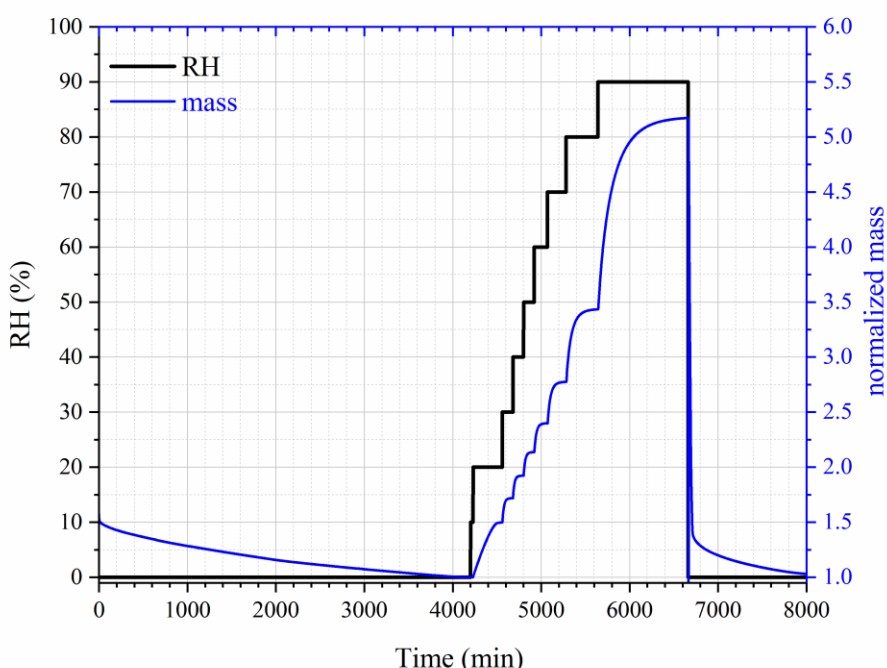

**Figure 3.** Change of normalized sample mass (blue curve, right *y*-axis) and RH (black curve, left *y*-axis) as a function of time for $CaCl_2·xH_2O$ at 25 °C.

Assuming that $CaCl_2 \cdot 6H_2O$ was converted to $CaCl_2 \cdot 2H_2O$ after being dried at 0% RH, we
could use the change of sample mass as a function of RH to calculate WSR (defined as molar ratio
of $H_2O$ to $Ca^{2+}$), and the results are listed in Table 3. Please note that we did not calculate WSR at
20% RH, since it is speculated that the significant mass increase at 20% RH was caused by the
transformation of $CaCl_2 \cdot 2H_2O$ to $CaCl_2 \cdot 6H_2O$, as mentioned above. Water activities of aqueous
$CaCl_2$ solutions as a function of molality concentration reported in a previous study (Rard et al.,
1977) were used to calculate WSR as a function of RH, and the results are also included in Table
3 for comparison. As evident from Table 3, at same/similar RH, WSR measured in our work are
in good agreement with those derived from Rard et al. (1977), supporting our assertion that
$CaCl_2 \cdot 6H_2O$ was converted to $CaCl_2 \cdot 2H_2O$ after being dried at 0% RH. In fact, theoretical
calculations (Kelly and Wexler, 2005) and experimental measurements (Gough et al., 2016) both
suggested that when RH is gradually increased, solid-solid phase transition from $CaCl_2 \cdot 2H_2O$ to
$CaCl_2 \cdot 6H_2O$ would occur before deliquescence takes place.

**Table 3.** Mass growth factors ($m/m_0$, defined as the ratio of sample mass at a given RH to that at
0% RH) and water-to-solute ratios (WSR) as a function of RH (0-90%) at 25 $^oC$ for $CaCl_2 \cdot xH_2O$.
WSR derived from RH over aqueous $CaCl_2$ solutions as a function of concentration (mol $kg^{-1}$) at
25 $^oC$ (Rard et al., 1977) are also included for comparison. All the errors given in this work are
standard deviations.

| our work | | | Rard et al., 1977 | | |
|---|---|---|---|---|---|
| RH (%) | $m/m_0$ | WSR | RH (%) | molality | WSR |
| 0 | 1.000±0.001 | -- | -- | -- | -- |
| 10 | 1.000±0.001 | -- | -- | -- | -- |
| 20 | 1.448±0.072 | -- | -- | -- | -- |
| 30 | 1.724±0.007 | 7.97±0.03 | 31.2 | 7.0 | 7.94 |

| | | | | | |
|---|---|---|---|---|---|
| 40 | 1.929±0.008 | 9.64±0.04 | 39.2 | 6.0 | 9.26 |
| 50 | 2.144±0.010 | 11.40±0.05 | 49.9 | 5.0 | 11.11 |
| 60 | 2.408±0.012 | 13.55±0.07 | -- | -- | -- |
| 70 | 2.786±0.015 | 16.64±0.09 | 70.1 | 3.4 | 16.34 |
| 80 | 3.448±0.020 | 22.05±0.13 | 79.8 | 2.6 | 21.37 |
| 90 | 5.194±0.030 | 36.30±0.21 | 89.9 | 1.6 | 37.72 |

Additional experiments, in which RH was stepwise increased from 0% with an increment of 1% per step, were carried out in attempt to measure the DRH of $CaCl_2 \cdot xH_2O$ at 25 °C. In all of these experiments, $CaCl_2 \cdot 6H_2O$ was always transformed to $CaCl_2 \cdot 2H_2O$ after being dried at 0% RH. In some of these experiments the deliquescence took place at RH of ~28.5%, which is consistent with the DRH of $CaCl_2 \cdot 6H_2O$ reported in the literature (Kelly and Wexler, 2005), suggesting that $CaCl_2 \cdot 2H_2O$ was first transformed to $CaCl_2 \cdot 6H_2O$ prior to deliquescence. However, in some other experiments the deliquescence occurred at RH of ~18.5%, corresponding to the DRH of $CaCl_2 \cdot 2H_2O$ reported previously (Kelly and Wexler, 2005), implying that $CaCl_2 \cdot 2H_2O$ was deliquesced without being transformed to $CaCl_2 \cdot 6H_2O$. The dual deliquescence processes, i.e. 1) transformation of $CaCl_2 \cdot 2H_2O$ to $CaCl_2 \cdot 6H_2O$ prior to deliquescence and 2) direct deliquescence of $CaCl_2 \cdot 2H_2O$, were also observed using Raman spectroscopy at low temperatures (223-273 K) (Gough et al., 2016). It seems that the competition of these two mechanisms are both thermodynamically and kinetically dependent. Since phase transitions of $CaCl_2$ are not only important for atmospheric aerosols but may also play a role in the existence of liquid water in some hyperarid environments (Gough et al., 2016), further investigation is being carried out by combining the vapor sorption analyzer technique with vibrational spectroscopy.

**3.1.4 Hygroscopic growth of aerosol particles**

Hygroscopic growth factors (GF), which were measured using H-TDMA at room
temperature, are displayed in Figure 4 for $Ca(NO_3)_2$, $CaCl_2$, $Mg(NO_3)_2$ and $MgCl_2$ aerosols, and
the results are also compiled in Table 4. It was found in our work that all the four types of aerosols
exhibit high hygroscopicity, with GF at 90% RH being around 1.7 or larger. In addition, all the
four types of aerosol particles, instead of having distinct solid-liquid phase transitions, showed
significant hygroscopic growth at very low RH (as low as 10%), and their GF increased
continuously with RH. This phenomenon is due to the fact that these aerosol particles, generated
by drying aqueous droplets, were likely to be amorphous. It was also observed in previous work
that some types of particles generated by drying aqueous droplets would be amorphous, such as
$Ca(NO_3)_2$ (Tang and Fung, 1997; Gibson et al., 2006; Jing et al., 2018), $Mg(NO_3)_2$ (Zhang et al.,
2004; Gibson et al., 2006; Li et al., 2008a), $CaCl_2$ (Park et al., 2009; Tobo et al., 2009) and $MgCl_2$
(Cziczo and Abbatt, 2000; Park et al., 2009).

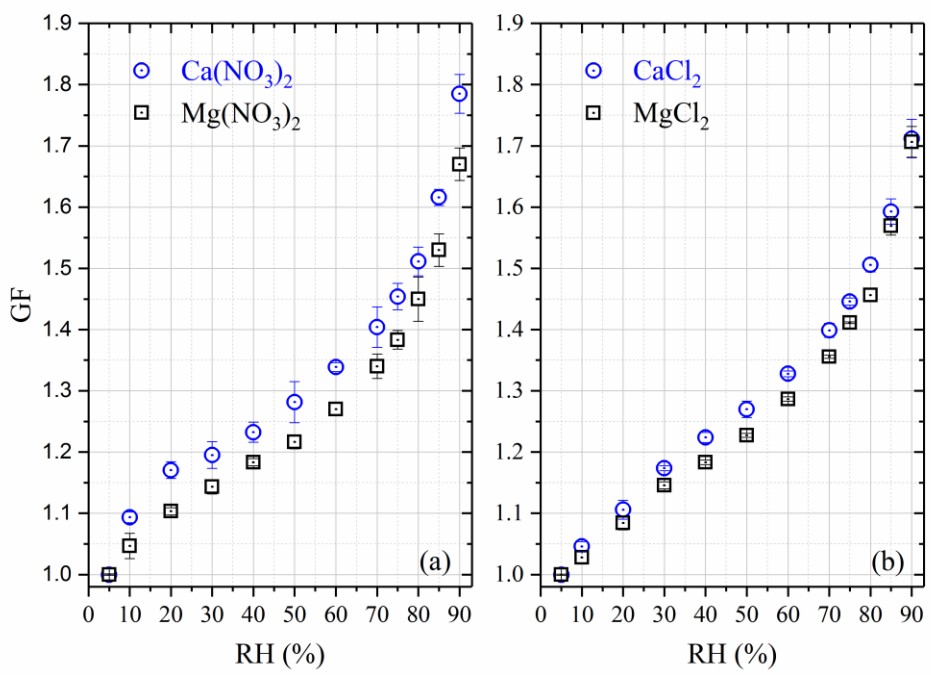


**Figure 4.** Hygroscopic growth factors (GF) of aerosol particles as a function of RH measured
using H-TDMA. (a): $Ca(NO_3)_2$ and $Mg(NO_3)_2$; (b) $CaCl_2$ and $MgCl_2$.

**$Ca(NO_3)_2$ and $Mg(NO_3)_2$ aerosols:** Two previous studies (Gibson et al., 2006; Jing et al.,

2018) employed H-TDMA to examine hygroscopic growth of 100 nm $Ca(NO_3)_2$ aerosol particles
at room temperature. GF were determined to be 1.51 at 80% RH and ~1.77 at 85% RH by Gibson
et al. (2008). It should be pointed out that though the DMA-selected dry particle diameters were
100 nm for $Ca(NO_3)_2$ and $Mg(NO_3)_2$ aerosols, the dry diameters used by Gibson et al. (2006) were
89 nm for $Ca(NO_3)_2$ and 77 nm for $Mg(NO_3)_2$, being extrapolated to 0% RH using the theoretical
growth curve based on the Köhler theory. The Köhler theory is based on assumption of solution
ideality, and thus may not be applicable to highly concentrated aerosol droplets at low RH
(Seinfeld and Pandis, 2016). If the dry diameter selected using the DMA (i.e. 100 nm) was used in
GF calculation, GF reported by Gibson et al. (2006) would be ~1.34 at 80% RH and ~1.58 at 85%
RH; compared with our results (1.51±0.02 at 80% RH and 1.62±0.01 at 85% RH), GF reported by
Gibson et al. (2006) are ~11% smaller at 80% RH and only ~3% smaller at 85%. In the second
study (Jing et al., 2018), GF were determined to be 1.56 at 80% RH and 1.89 at 90% RH; compared
with our results (1.51±0.02 at 80% RH and 1.79±0.03 at 90% RH), GF reported by Jing et al.
(2018) were ~3% larger at 80% RH and ~6% larger at 90% RH. Overall, our results show
reasonably good agreement with the two previous studies (Gibson et al., 2006; Jing et al., 2018).

**Table 4.** Hygroscopic growth factors (GF) of $Ca(NO_3)_2$, $CaCl_2$, $Mg(NO_3)_2$ and $MgCl_2$ aerosol
particles measured at room temperature using H-TDMA. The absolute uncertainties in RH were
estimated to be within ±2%. All the errors given in this work are standard deviations.

| RH (%) | $Ca(NO_3)_2$ | $CaCl_2$ | $Mg(NO_3)_2$ | $MgCl_2$ |
|---|---|---|---|---|
| <5 | 1.00±0.01 | 1.00±0.01 | 1.00±0.01 | 1.00±0.01 |
| 10 | 1.09±0.01 | 1.05±0.01 | 1.05±0.02 | 1.03±0.01 |

| | | | |
|---|---|---|---|
| 20 | 1.17±0.02 | 1.11±0.02 | 1.10±0.01 | 1.08±0.01 |
| 30 | 1.20±0.02 | 1.17±0.01 | 1.41±0.01 | 1.15±0.01 |
| 40 | 1.23±0.02 | 1.22±0.01 | 1.18±0.01 | 1.18±0.01 |
| 50 | 1.28±0.03 | 1.27±0.01 | 1.22±0.01 | 1.23±0.01 |
| 60 | 1.34±0.01 | 1.33±0.01 | 1.27±0.01 | 1.29±0.01 |
| 70 | 1.40±0.03 | 1.40±0.01 | 1.34±0.02 | 1.36±0.01 |
| 75 | 1.45±0.02 | 1.45±0.01 | 1.38±0.02 | 1.41±0.01 |
| 80 | 1.51±0.02 | 1.51±0.01 | 1.45±0.04 | 1.46±0.01 |
| 85 | 1.62±0.01 | 1.59±0.02 | 1.53±0.03 | 1.57±0.02 |
| 90 | 1.79±0.03 | 1.71±0.03 | 1.67±0.03 | 1.71±0.03 |


To our knowledge, only one previous study investigated the hygroscopic growth of $Mg(NO_3)_2$ aerosol (100 nm) using the H-TDMA (Gibson et al., 2006), and GF was measured to be 1.94±0.02 at 83% RH. As stated above, the theoretical extrapolated diameter (77 nm) at 0% RH, instead of the dry diameter (100 nm) selected using the DMA, were used as the dry diameter to calculate their reported GF (Gibson et al., 2006). If the DMA-selected dry diameter (100 nm) was used in calculation, the GF reported by Gibson et al. (2006) would be ~1.49 at 83% RH; for comparison, in our work GF were determined to be 1.45±0.04 and 1.53±0.03 at 80 and 85% RH, suggesting good agreement between the two studies if the DMA-selected dry diameter was used to calculate GF reported by Gibson et al. (2006).

**$CaCl_2$ and $MgCl_2$ aerosols:** Hygroscopic growth of $CaCl_2$ and $MgCl_2$ aerosol particles was explored using a H-TDMA (Park et al., 2009), and as far as we know, this was the only study which reported the H-TDMA measured hygroscopic growth factors of the two types of aerosols. Three dry diameters (20, 30 and 50 nm) were used for $CaCl_2$ and $MgCl_2$ aerosol particles (Park et al., 2009), and no significant size dependence of their hygroscopic properties was observed. GF were measured to be around 1.27, 1.38, 1.48 and 1.59 at 60, 75, 80 and 90 % RH for $CaCl_2$ (Park

et al., 2009). For comparison, GF were determined in this work to be 1.33±0.01, 1.45±0.01,
1.51±0.01 and 1.71±0.03 at 60, 75, 80 and 90 %, slightly larger than those reported by Park et al.
(2009), and the differences were found to be <7%.
At 50, 70, 80, 85 and 90% RH, GF of $MgCl_2$ aerosol were measured to be about 1.17, 1.29,
1.47, 1.59 and 1.79 by Park et al. (2009); for comparison, GF were determined to be 1.23±0.01,
1.36±0.01, 1.46±0.01, 1.57±0.02 and 1.71±0.03 in our work at the same RHs. The differences did
not exceed 6% at any given RH, suggesting good agreement between the two studies. Microscopy
was used to investigate the hygroscopic growth of micrometer-size $MgCl_2$ particles deposited on
substrates (Gupta et al., 2015), and the ratios of 2-D particle areas, relative to that at <5% RH,
were measured to be around 1.65, 1.92, 2.02 and 2.28 at 60, 70, 75 and 80% RH, corresponding to
diameter-based GF of approximately 1.28, 1.38, 1.42 and 1.51, respectively. GF of $MgCl_2$ aerosol,
as shown in Table 4, were determined to be 1.29±0.01, 1.36±0.01, 1.41±0.01 and 1.46±0.01 at 60,
70, 75 and 80% RH in our work; therefore, the differences between GF reported in our work and
those measured by Gupta et al. (2015) were <4%.
**Comparison between hygroscopic growth with CCN activities:** GF measured using H-
TDMA can be used to calculate the single hygroscopicity parameter, $\kappa_{gf}$, using Eq. (3a) (Petters
and Kreidenweis, 2007; Kreidenweis and Asa-Awuku, 2014; Tang et al., 2016a):
$$\frac{RH}{\exp(\frac{A_k}{d_0 \cdot GF})} = \frac{GF^3 - 1}{GF^3 - (1 - \kappa_{gf})} \qquad (3a)$$
where GF is the growth factor at a given RH; $A_k$ is a constant which describes the Kelvin effect
and is equal to 2.1 nm for a surface tension of 0.072 J m$^{-2}$ (pure water) and temperature of 298.15
K (Tang et al., 2016a). For a dry particle diameter ($d_0$) of 100 nm, the denominator in the left term
of Eq. (3a) is not larger than 1.02; therefore, the Kelvin effect is negligible and Eq. (3a) can be
simplified to Eq. (3b):
$$\text{RH} = \frac{GF^3 - 1}{GF^3 - (1 - \kappa_{gf})} \qquad (3b)$$
Eq. (4) can be derived by rearranging Eq. (3b):
$$\kappa_{gf} = (GF^3 - 1)\frac{1 - RH}{RH} \qquad (4)$$
In our work, GF data at 90% RH were used to derive $\kappa_{gf}$, as usually done in many previous studies
(Kreidenweis and Asa-Awuku, 2014). The single hygroscopicity parameter, $\kappa_{ccn}$, can also be
derived from experimental measurements or theoretical calculations of CCN activities (Petters and
Kreidenweis, 2007; Kreidenweis and Asa-Awuku, 2014). Ideally aerosol-water interactions under
both subsaturation and supersaturation can be described by a constant single hygroscopicity
parameter (Petters and Kreidenweis, 2007). Nevertheless, agreement and discrepancies between
growth factors derived and CCN activity derived $\kappa$ have been reported (Petters and Kreidenweis,
2007; Petters et al., 2009; Wex et al., 2009), and several factors can contribute to such
discrepancies. First of all, the solutions may not be ideal, and especially aerosol particles under
subsaturation may consist of concentrated solutions; secondly, some of the compounds may have
limited solubilities. As discussed previously (Petters and Kreidenweis, 2007; Prenni et al., 2007),
both factors would lead to lower $\kappa_{gf}$, compared to $\kappa_{ccn}$. The effect of reduced surface tension,
compared to pure water, should be negligible for the eight types of aerosol particles considered in
our work, since none of these compounds are known to be surface-active.
Comparison between $\kappa_{gf}$ determined in our work and $\kappa_{ccn}$ measured in previous studies is
summarized in Table 5 and discussed below for $Ca(NO_3)_2$, $CaCl_2$, $Mg(NO_3)_2$ and $MgCl_2$ aerosols.
In previous work which measured CCN activities (Sullivan et al., 2009; Tang et al., 2015; Gaston
et al., 2017), the dry particle diameters used were typically in the range of 50-125 nm. The
uncertainties in our derived $\kappa_{gf}$ have taken into account the uncertainties in measured GF at 90%
RH.

**Table 5.** Comparison between $\kappa_{gf}$ measured in our work and $\kappa_{ccn}$ measured in previous studies.

| aerosol | $\kappa_{gf}$ (this work) | $\kappa_{ccn}$ (previous studies) |
|---|---|---|
| $Ca(NO_3)_2$ | 0.49-0.56 | 0.44-0.64 (Sullivan et al., 2009) |
| | | 0.57-0.59 (Tang et al., 2015) |
| $Mg(NO_3)_2$ | 0.38-0.43 | not measured yet |
| $CaCl_2$ | 0.42-0.47 | 0.46-0.58 (Sullivan et al, 2009) |
| | | 0.51-0.54 (Tang et al, 2015)) |
| | | 0.549-0.561 (Gaston et al., 2017) |
| $MgCl_2$ | 0.42-0.47 | 0.456-0.464 (Gaston et al., 2017) |
| $Ca(HCOO)_2$ | 0.28-0.31 | 0.47-0.52 (Tang et al., 2015) |
| $Mg(HCOO)_2$ | 0.40-0.45 | not measured yet |
| $Ca(CH_3COO)_2$ | 0.09-0.13 | 0.37-0.47 (Tang et al., 2015) |
| $Mg(CH_3COO)_2$ | 0.28-0.29 | not measured yet |


1) For $Ca(NO_3)_2$ aerosol, $\kappa_{ccn}$ were measured to be 0.44-0.64 by Sullivan et al. (2009) and

0.57-0.59 by Tang et al. (2015); in our work GF at 90% RH was measured to be 1.79±0.03, giving
$\kappa_{gf}$ of 0.49-0.56, in good agreement with $\kappa_{ccn}$ reported by the two previous studies (Sullivan et al.,
2009; Tang et al., 2015).

2) For $CaCl_2$ aerosol, $\kappa_{ccn}$ were measured to be 0.46-0.58 by Sullivan et al. (2009), 0.51-

0.54 by Tang et al. (2015), and 0.549-0.561 by Gaston et al. (2017). GF at 90% RH was determined
to be 1.71±0.03 in present work, giving $\kappa_{gf}$ of 0.42-0.47, slightly lower than $\kappa_{ccn}$ values measured
previously (Sullivan et al., 2009; Tang et al., 2015; Gaston et al., 2017).
3) In our work, GF was determined to be 1.71±0.03 for $MgCl_2$ at 90% RH, giving $\kappa_{gf}$ of
0.42-0.47; a previous study (Gaston et al., 2017) measured the CCN activity of $MgCl_2$ aerosol, and
$\kappa_{ccn}$ were determined to be 0.456-0.464, in good agreement with $\kappa_{gf}$ measured in our work.
4) For $Mg(NO_3)_2$ aerosol, GF and $\kappa_{gf}$ were determined in our work to be 1.67±0.03 and
0.38-0.43, respectively. To our knowledge, CCN activities of $Mg(NO_3)_2$ aerosol have not been
experimentally explored yet, and $\kappa_{ccn}$ were predicted to be 0.8 for $Mg(NO_3)_2$ and 0.3 for
$Mg(NO_3)_2 \cdot 6H_2O$ (Kelly et al., 2007; Kreidenweis and Asa-Awuku, 2014), exhibiting a large
variation for the same compound with different hydrate states under dry conditions. These
calculations were performed using the Köhler theory, assuming solution ideality (Kelly et al.,
2007). As Kelly et al. (2007) pointed out, the hydration states, which are not entirely clear for
$Mg(NO_3)_2$ aerosol particles under atmospherically relevant conditions, can have large impacts on
their hygroscopicity and CCN activities.
**3.2 Hygroscopicity of formates and acetates**
**3.2.1 DRH and water-to-solute ratios**
We measured the mass change of $Ca(HCOO)_2$, $Mg(HCOO)_2 \cdot 2H_2O$ and
$Ca(CH_3COO)_2 \cdot H_2O$ samples as a function of RH at 25 °C, and found that the sample mass
remained essentially constant for all the three compounds when RH was increased from 0 to 90%.
Therefore, a series of experiments in which RH was increased to 95% were conducted, and for
each compounds three duplicate experiments were carried out. As shown in Figure 5a, when RH
was increased from 0 to 95%, a significant while small increase in sample mass (~10%) was
observed for $Ca(HCOO)_2$. The average ratio of sample mass at 95% RH to that at 0% RH was
determined to be for 1.119±0.036 for $Ca(HCOO)_2$ and 1.064±0.020 for $Mg(HCOO)_2 \cdot 2H_2O$ (not

shown in Figure 5), probably indicating that the DRH values were >95% for both compounds at 25 °C.

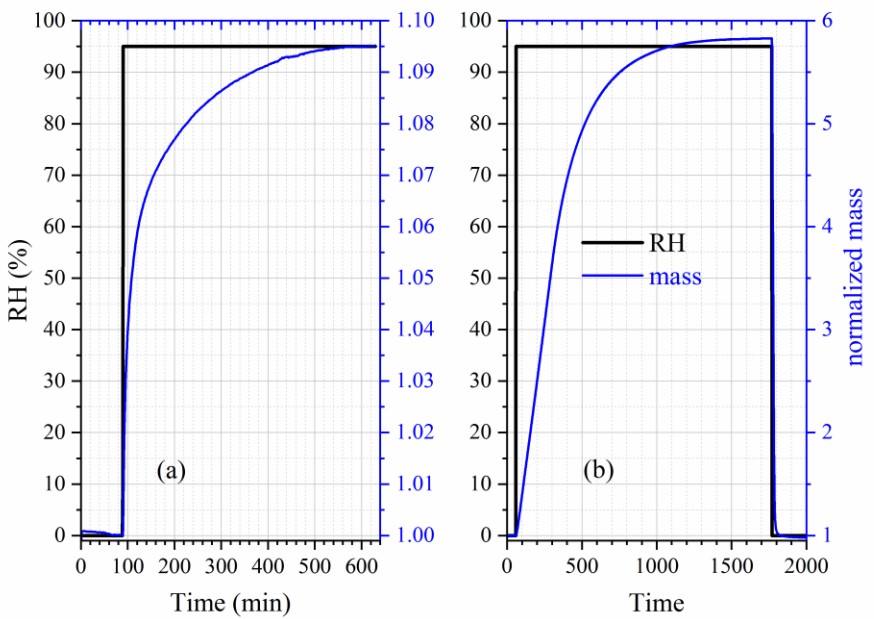

**Figure 5.** Change of normalized sample mass (blue curve, right *y*-axis) and RH (black curve, left *y*-axis) as a function of time at 25 °C. (a) $Ca(HCOO)_2$; (b) $Ca(CH_3COO)_2 \cdot H_2O$.

When RH was increased from 0 to 95%, large increase in sample mass (almost by a factor of 6), as shown in Figure 6b, was observed for $Ca(CH_3COO)_2 \cdot H_2O$. On average, the ratio of sample mass at 95% RH to that at 0% RH was measured to be 5.849±0.064, corresponding to a WSR (defined as the molar ratio of $H_2O$ to $Ca^{2+}$) of 48.42±0.53 for the aqueous $Ca(CH_3COO)_2$ solution at 95% RH. This observation suggested that the deliquescence of $Ca(CH_3COO)_2 \cdot H_2O$ at 25 °C occurred between 90 and 95% RH. In further experiments significant increase in sample mass (by >10%, and the sample was still increasing sharply when the experiment was terminated) was observed when RH was increased from 90 to 91% for $Ca(CH_3COO)_2 \cdot H_2O$ at 25 °C, suggesting a measured DRH of 90.5±1.0 %. The DRH of $Ca(CH_3COO)_2$ and internally mixed

$CaCO_3$/$Ca(CH_3COO)_2$ particles were measured to be 85 and 88% at 5 $^{\circ}$C (Ma et al., 2012), using
a modified physisorption analyzer. Since in these two studies DRH were measured at different
temperatures (25 $^{\circ}$C in our work and 5 $^{\circ}$C by Ma et al.) and the absolute difference in reported
DRH was ~5%, the agreement in reported DRH can be considered to be quite good for
$Ca(CH_3COO)_2$.
Table 6 summarizes the ratios of sample mass at a given RH to that at 0% RH for
$Mg(CH_3COO)_2 \cdot 4H_2O$ as a function of RH at 25$^{\circ}$C. Being different from $Ca(HCOO)_2$,
$Mg(HCOO)_2 \cdot 2H_2O$ and $Ca(CH_3COO)_2 \cdot H_2O$, large increase in sample mass was observed for
$Mg(CH_3COO)_2 \cdot 4H_2O$ when RH was increased from 70 to 80%. This observation suggested that
the deliquescence of $Mg(CH_3COO)_2 \cdot 4H_2O$ occurred between 70 and 80% RH. Further
experiments were carried out to measure its DRH, and significant increase in sample mass occurred
when RH was increased from 71 to 72%, giving a measured DRH of 71.5±1.0% at 25 $^{\circ}$C. The RH
over the saturated $Mg(CH_3COO_2)_2$ solution at ~23 $^{\circ}$C was measured to be 65% (Wang et al., 2005),
slightly lower than the DRH determined in our work.

**Table 6.** Mass growth factors ($m/m_0$, defined as the ratios of sample mass at a given RH to that at
0% RH) and water-to-solute ratios (WSR) as a function of RH (0-90%) at 25 $^{\circ}$C for
$Mg(CH_3COO)_2 \cdot 4H_2O$. WSR are only calculated for RH exceeding the DRH (i.e. when the sample
was deliquesced). All the errors given in this work are standard deviations.

| RH (%) | 0 | 10 | 20 | 30 | 40 |
|---|---|---|---|---|---|
| $m/m_0$ | 1.000±0.001 | 1.012±0.021 | 1.012 ±0.022 | 1.013 ±0.022 | 1.013±0.022 |
| WSR | -- | -- | -- | -- | -- |
| RH (%) | 50 | 60 | 70 | 80 | 90 |
| $m/m_0$ | 1.014±0.023 | 1.015±0.025 | 1.033±0.031 | 2.029±0.013 | 3.100±0.021 |
| WSR | -- | -- | -- | 16.24±0.11 | 28.97±0.20 |


The ratios of sample mass, relative to that at 0% RH, were measured to be 2.029±0.013 and 3.100±0.021 at 80 and 90% RH, corresponding to WSR of 16.24±0.11 at 80% RH and 28.97±0.20 at 90% RH for aqueous $Mg(CH_3COO)_2$ solutions. A electrodynamic balance coupled to Raman spectroscopy was employed to study the hygroscopic growth of $Mg(CH_3COO)_2$ at ~23 °C (Wang et al., 2005), and WSR was determined to be ~15.6 at 80% RH, in good agreement with our work. Ma et al. (2012) found that after heterogeneous reaction with $CH_3COOH(g)$ at 50% RH for 12 h, the hygroscopicity of MgO particles, which was initially rather non-hygroscopic, was substantially increased due to the formation of $Mg(CH_3COO)_2$. The conclusion drawn by Ma et al. (2012) is qualitatively consistent with the results obtained in our work.

Table 6 also reveals that a small increase in sample mass (by ~3%, relative to that at 0% RH) was observed for $Mg(CH_3COO)_2·4H_2O$ when RH was increased to 70% before the deliquescence of $Mg(CH_3COO)_2·4H_2O$ took place. This could be due to the possibility that $Mg(CH_3COO)_2·4H_2O$ samples used in our work may contain a small fraction of amorphous $Mg(CH_3COO)_2$, which would take up some amount of water at RH below the DRH of $Mg(CH_3COO)_2·4H_2O$ (Wang et al., 2005; Pang et al., 2015).

**3.2.2 Hygroscopic growth of aerosol particles**

Figure 6 and Table 7 display hygroscopic growth factors of $Ca(HCOO)_2$, $Mg(HCOO)_2$, $Ca(CH_3COO)_2$ and $Mg(CH_3COO)_2$ aerosols, measured in our work using H-TDMA. To the best of our knowledge, this is the first time that GF of these four types of aerosols have been reported. For $Mg(HCOO)_2$, aerosol particles showed gradual while small growth for RH up to 30%, and further increase in RH led to significant growth; the average GF of $Mg(HCOO)_2$ aerosol at 90% RH was determined to be 1.69±0.03, similar to those for $Mg(NO_3)_2$ (1.67±0.03) and $MgCl_2$

(1.71±0.03) at the same RH. For RH up to 85%, $Ca(HCOO)_2$ aerosol particles exhibited gradual
and small growth; when RH was increased to 90%, abrupt and large growth was observed, with
GF being 1.54±0.02, significantly smaller than that for $Mg(HCOO)_2$ aerosol at the same RH. This
is distinctively different from what was observed in VSA experiments, in which the mass of
$Ca(HCOO)_2$ and $Mg(HCOO)_2 \cdot 2H_2O$ powdered samples was only increased by ~12% and ~6%
when RH was increased from 0 to 95%. This difference may be explained by different states of
samples used in these two types of experiments (i. e. crystalline samples in VSA experiments,
while likely amorphous aerosol particles in H-TDMA measurements), leading to different
hygroscopic behaviors.

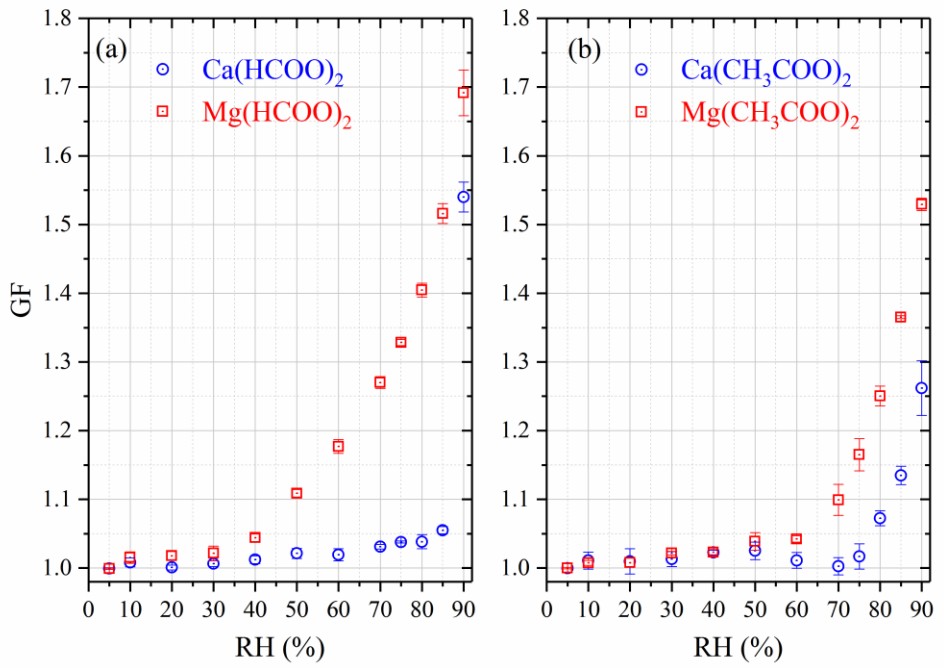


**Figure 6.** Hygroscopic growth factors (GF) of aerosol particles as a function of RH measured
using H-TDMA. (a): $Ca(HCOO)_2$ and $Mg(HCOO)_2$; (b) $Ca(CH_3COO)_2$ and $Mg(CH_3COO)_2$.

As shown in Figure 6b, gradual and small growth was also observed for $Ca(CH_3COO)_2$
and $Mg(CH_3COO)_2$ aerosols at low RH. Fast increase in GF started at about 80% RH for
$Ca(CH_3COO)_2$ aerosol, and the GF was determined to be 1.26±0.04 at 90% RH. As discussed in
Section 3.2.1, in VSA experiments no significant increase in sample mass was observed for
$Ca(CH_3COO)_2 \cdot H_2O$ when RH was increased from 0 to 90%, being different from H-TDMA results.
This difference may again be explained (at least partly) by different states of particles used in these
two types of experiments, as mentioned above. Careful inspection of Figure 6b and Table 7 reveals
that a small decrease in GF from 1.03±0.01 to 1.00±0.01 for $Ca(CH_3COO)_2$ aerosol when RH was
increased from 50 to 70%. The decrease in GF may be caused by restructuring of particles or
change in particle morphology (Vlasenko et al., 2005; Koehler et al., 2009); in addition, the small
change in GF (~0.03) may not be significant when compared to the uncertainties in our H-TDMA
measurements.
When RH increased from 0 to 70%, small and gradual growth occurred for $Mg(CH_3COO)_2$
aerosol particles, indicating that these particles may contain some amount of amorphous materials.
It was also found in previous work (Li et al., 2008a; Li et al., 2008b) that $Mg(NO_3)_2$ particles
generated by drying aqueous droplets were amorphous. Figure 6b reveals that further increase in
RH led to large increase in growth factors, and this is largely consistent with the occurrence of
deliquescence at ~71.5% RH at 25 °C for $Mg(CH_3COO)_2 \cdot 4H_2O$, as mentioned in Section 3.2.1. At
90% RH, GF of $Mg(CH_3COO)_2$ aerosol was determined to be 1.53±0.01, much larger than that for
$Ca(CH_3COO)_2$ (1.26±0.04).
At 90% RH, for the four Ca-containing salts considered in our study, nitrate and chloride
aerosols have very similar GF (1.79±0.03 versus 1.71±0.03), which are large than that of formate
(1.54±0.02), and acetate has the smallest GF (1.26±0.04). For comparison, the variation in GF at
90% RH was found to be considerably smaller (from ~1.53 to ~1.71) for the four Mg-containing
salts studied herein.

**Table 7.** Hygroscopic growth factors of $Ca(HCOO)_2$, $Ca(CH_3COO)_2$, $Mg(HCOO)_2$ and

$Mg(CH_3COO)_2$ aerosol particles measured using H-TDMA. The absolute uncertainties in RH were

estimated to be within ±2%. All the errors given in this work are standard deviations.

| RH (%) | $Ca(HCOO)_2$ | $Ca(CH_3COO)_2$ | $Mg(HCOO)_2$ | $Mg(CH_3COO)_2$ |
|---|---|---|---|---|
| 5 | 1.00±0.01 | 1.00±0.01 | 1.00±0.01 | 1.00±0.01 |
| 10 | 1.01±0.01 | 1.01±0.01 | 1.02±0.01 | 1.01±0.01 |
| 20 | 1.01±0.01 | 1.01±0.02 | 1.02±0.01 | 1.01±0.01 |
| 30 | 1.01±0.01 | 1.01±0.01 | 1.02±0.01 | 1.02±0.01 |
| 40 | 1.01±0.01 | 1.02±0.01 | 1.04±0.01 | 1.02±0.01 |
| 50 | 1.02±0.01 | 1.03±0.01 | 1.11±0.01 | 1.04±0.01 |
| 60 | 1.02±0.01 | 1.01±0.01 | 1.18±0.01 | 1.04±0.01 |
| 70 | 1.03±0.01 | 1.00±0.01 | 1.27±0.01 | 1.10±0.02 |
| 75 | 1.04±0.01 | 1.02±0.02 | 1.33±0.01 | 1.16±0.02 |
| 80 | 1.04±0.01 | 1.07±0.01 | 1.41±0.01 | 1.25±0.01 |
| 85 | 1.01±0.01 | 1.13±0.01 | 1.52±0.02 | 1.37±0.01 |
| 90 | 1.54±0.02 | 1.26±0.04 | 1.69±0.03 | 1.53±0.01 |


According to Eq. (4), GF measured at 90% RH can be used to calculate $\kappa_{gf}$, which were

determined to be 0.28-0.31 for $Ca(HCOO)_2$, 0.09-0.13 for $Ca(CH_3COO)_2$, 0.40-0.45 for

$Mg(HCOO)_2$, and 0.28-0.29 for $Mg(CH_3COO)_2$. A previous study (Tang et al., 2015) investigated

the CCN activities of $Ca(HCOO)_2$ and $Ca(CH_3COO)_2$ aerosols and reported their single

hygroscopicity parameters ($\kappa_{ccn}$), while the CCN activities of $Mg(HCOO)_2$ and $Mg(CH_3COO)_2$

have not been explored yet. As summarized in Table 5, $\kappa_{ccn}$ was reported to be 0.47-0.52 for

$Ca(HCOO)_2$ (Tang et al., 2015), significantly larger than $\kappa_{gf}$ (0.28-0.31) determined in our work;

for $Ca(CH_3COO)_2$, Tang et al. (2015) reported $\kappa_{ccn}$ to be in the range of 0.37-0.47, again much

larger than $\kappa_{gf}$ (0.09-0.13) derived from the present work.

As discussed in Section 3.1.4, for $Ca(NO_3)_2$ and $CaCl_2$ aerosols, $\kappa_{gf}$ derived from H-TDMA
experiments in the present work show fairly good agreement with $\kappa_{ccn}$ derived from CCN activities
measured in previous studies (Sullivan et al., 2009; Tang et al., 2015); in contrast, for $Ca(HCOO)_2$
and $Ca(CH_3COO)_2$ aerosols, $\kappa_{gf}$ derived from our H-TDMA experiments are significantly smaller
than $\kappa_{ccn}$ reported by the previous study (Tang et al., 2015). This can be largely caused by the
difference in water solubilities of $Ca(NO_3)_2$, $CaCl_2$, $Ca(HCOO)_2$ and $Ca(CH_3COO)_2$.
$Ca(NO_3)_2 \cdot 4H_2O$ and $CaCl_2 \cdot 6H_2O$, with solubilities being 1983 and 1597 g per kg water at 25 $^o$C
(Kelly and Wexler, 2005), can be considered to be highly soluble; for comparison, the solubilities
were reported to be 166 g per kg water for $Ca(HCOO)_2$ at 25 $^o$C and 347 g per kg water for
$Ca(CH_3COO)_2 \cdot 2H_2O$ at 20 $^o$C (Dean, 1973). Due to their limited water solubilities, $Ca(HCOO)_2$
and $Ca(CH_3COO)_2$ aerosol particles may not be fully dissolved at 90% RH in the H-TDMA
experiments but would be dissolved to a larger extent (if not completely) for RH >100% in CCN
activity measurements (Petters and Kreidenweis, 2008; Kreidenweis and Asa-Awuku, 2014).
Therefore, for $Ca(HCOO)_2$ and $Ca(CH_3COO)_2$ aerosols, $\kappa_{gf}$ derived from H-TDMA measurements
would be smaller than $\kappa_{ccn}$ derived from CCN activity measurements. In fact, the observation that
$\kappa_{gf}$ appeared to be significantly smaller than $\kappa_{ccn}$, largely caused by limited water solubilities of
compounds under investigation, has been well documented in the literature for laboratory-
generated and ambient aerosol particles (Chang et al., 2007; Prenni et al., 2007; Wex et al., 2009;
Good et al., 2010; Massoli et al., 2010).
**3.3 Discussion**
**3.3.1 Comparison between H-TDMA and VSA measurements**
In this work two complementary techniques were employed to investigate hygroscopic
properties of Ca- and Mg-containing compounds. The mass change of bulk samples was measured
as a function of RH using VSA, and the change in aerosol diameter with RH was determined using
H-TDMA. Two major questions can be asked regarding the results obtained using the two different
techniques: 1) How can the two types of results be reconciled? 2) What is the atmospheric
relevance of each type of results? Below we use $Ca(NO_3)_2$ at room temperature as an example for
discussion, and similar conclusions can be drawn for the other seven compounds.

As presented in Section 3.1, at 25 °C the deliquescence of $Ca(NO_3)_2 \cdot 4H_2O$ took place at

52-53% RH. In contract, dry $Ca(NO_3)_2$ aerosol particles generated by atomizing aqueous solutions
were likely to be amorphous (Tang and Fung, 1997; Al-Abadleh et al., 2003; Gibson et al., 2006);
as a result, they exhibited continuous hygroscopic growth with increasing RH with no distinct
solid-liquid phase transitions observed. When RH exceed the DRH of $Ca(NO_3)_2 \cdot 4H_2O$, both
$Ca(NO_3)_2 \cdot 4H_2O$ bulk samples and $Ca(NO_3)_2$ aerosol particles are expected to deliquesce to form
aqueous solutions. To directly link the mass change (measured using VSA) with diameter change
(measured using H-TDMA), solution densities, which also vary with RH, are needed. Two
important outputs of common aerosol thermodynamic models, such as E-AIM (Clegg et al., 1998)
and ISORROPIA II (Fountoukis and Nenes, 2007) are volumes and water-to-solute ratios as a
function of RH (above DRH) for aqueous solutions. Water-to-solute ratios and particle diameters
were both measured in our work at different RH, and our experimental data, when compared with
theoretical calculations, can be used to validate these thermodynamic models.

When RH are lower than the DRH of $Ca(NO_3)_2 \cdot 4H_2O$, aerosol particles used in our H-

TDMA experiments, instead of bulk samples used in the VSA measurements, are of direct
atmospheric relevance, and hence the H-TDMA results should be used in atmospheric applications.
There are still some open questions regarding $Ca(NO_3)_2$ aerosol particles (as well as other types of
particles investigated in this work) for RH below DRH of $Ca(NO_3)_2 \cdot 4H_2O$. What is the phase state
of aerosol particles at different RH? Are they crystalline solid, amorphous solid (glassy), or
supersaturated solutions? In this aspect, measurements of particle phase state of $Ca(NO_3)_2$ and
other aerosols considered in our work, using the apparatus described previously (Li et al., 2017),
can shed some light. Furthermore, how do water-to-solute ratios change with RH for $Ca(NO_3)_2$
aerosol particles when RH is below the DRH of $Ca(NO_3)_2 \cdot 4H_2O$? This can be answered by
determining particle mass as a function of RH for aerosol particles, and techniques are now
available for this task (Vlasenko et al., 2017).

## 682 3.3.2 Atmospheric implications

Hygroscopicity of carbonate minerals, such as calcite and dolomite, is initially very low
and can be largely enhanced due to formation of more hygroscopic materials via heterogeneous
reactions during transport (Tang et al., 2016a). Our present work investigated the hygroscopic
properties of eight Ca- or Mg-containing compounds which are aging products formed via
heterogeneous reactions of carbonate minerals, and revealed that the hygroscopicity of these
products is significantly higher than original carbonate minerals. In addition, hygroscopicity was
found to differ for different aging products, suggesting that heterogeneous reactions with different
trace gases may have distinctive effects on the hygroscopicity of carbonate minerals. For example,
the hygroscopicity of $Ca(NO_3)_2$ and $CaCl_2$, formed through heterogeneous reactions with nitrogen
oxides and HCl, is much higher than that for $Ca(HCOO)_2$ and $Ca(CH_3COO)_2$, formed via
heterogeneous reactions with formic and acidic acids. Our work also observed that significant
hygroscopic growth of aerosol particles, such as $Ca(NO_3)_2$ and $CaCl_2$, occurred at RH as low as
10%. This implies that aged carbonate particles can take up significant amount of water even under
very low RH, leading to changes in their diameters and morphology and thus impacting their
optical properties and direct radiative effects (Pan et al., 2015; Pan et al., 2018).

698    Large amounts of saline mineral dust are emitted into the atmosphere from dry lake beds

699 (Prospero et al., 2002), but these particles are usually assumed to be nonhygroscopic. Gaston et al.

700 (2017) found that saline mineral dust particles from different sources exhibit very different CCN

701 activities, and the measured $\kappa_{ccn}$ varied from <0.01 to >0.8, depending on the abundance of soluble

702 components (e.g., chlorides and sulfates) contained in these particles. Saline mineral dust particles

703 from different sources are very likely to have different hygroscopic properties under subsaturation.

704 To understand the hygroscopic growth of saline mineral dust particles, knowledge in hygroscopic

705 growth as well as the abundance of soluble components they contain is needed. Since $CaCl_2$ and

706 $MgCl_2$ have been identified as important components in saline mineral dust, their hygroscopicity

707 data measured in our work will be useful for improving our knowledge in hygroscopic properties

708 of saline mineral dust.

709    It is conventionally assumed that the hygroscopicity of sea salt is very similar to that of

710 pure NaCl. However, a recent study (Zieger et al., 2017) suggested that the hygroscopic growth

711 factor of sea salt aerosol at 90% RH is 8-15% lower than NaCl aerosol, and this difference is

712 attributed to the presence of $MgCl_2$ and $CaCl_2$ hydrates in sea salt. Growth factors at 90% RH were

713 measured in our work to be ~1.7 for $MgCl_2$ and $CaCl_2$ aerosols, significant lower that for NaCl

714 (2.29-2.46) (Zieger et al., 2017). Therefore, our work provides further experimental results to

715 support the conclusion drawn by Zieger et al. (2017), and would help better understand the

716 hygroscopicity of sea salt aerosol.

## 717 4. Summary and Conclusion

718    Ca- and Mg-containing salts, including nitrates, chlorides, formates and acetates, are

719 important components for mineral dust and sea salt aerosols; however, their hygroscopic properties

720 are not well understood yet. In this work, phase transition and hygroscopic growth of eight Ca- or

Mg-containing compounds were systematically examined using a vapor sorption analyzer and a humidity-tandem differential mobility analyzer. DRH values decreased from 60.5±1.0% at 5 °C to 46.0±1.0% at 30 °C for Ca(NO₃)₂·4H₂O and from 57.5±1.0% at 5 °C to 50.5±1.0% at 30 °C for Mg(NO₃)₂·6H₂O, both showing negative dependence on temperature, and this dependence can be approximated by the Clausius-Clapeyron equation. No significant dependence of DRH (around 31-33%) on temperature (5-30 °C) was observed for MgCl₂·6H₂O. CaCl₂·6H₂O, found to deliquesce at ~28.5% RH at 25 °C, exhibited complex phase transition processes in which CaCl₂·2H₂O, CaCl₂·6H₂O and aqueous CaCl₂ solutions were involved. Furthermore, DRH values were determined to be 90.5±1.0% for Ca(CH₃COO)₂·H₂O and 71.5±1.0% for Mg(CH₃COO)₂·4H₂O at 25 °C; for comparison, the sample mass was only increased by ~12% for Ca(HCOO)₂ and ~6% for Mg(HCOO)₂·2H₂O when RH was increased from 0 to 95%, implying that the DRH of these two compounds were probably >95%.

We have also measured the change of sample mass as a function of RH up to 90% to derive the water-to-solute ratios (WSR) for deliquesced samples. WSR were determined at 25 and 5 °C for deliquesced Ca(NO₃)₂·4H₂O, Mg(NO₃)₂·6H₂O and MgCl₂·6H₂O samples, and at 25 °C for deliquesced CaCl₂·6H₂O and Mg(CH₃COO)₂·4H₂O samples. We found that compared to that at 0% RH, large increases in sample mass only occurred when RH was increased from 90 to 95% for Ca(CH₃COO)₂·H₂O, and the WSR value was determined to be 5.849±0.064 at 95% RH. Besides, deliquescence was not observed even when RH was increased to 95% for Ca(HCOO)₂ and Mg(HCOO)₂·2H₂O, and the ratios of sample mass at 95% to that at 0% RH, were determined to be for 1.119±0.036 for Ca(HCOO)₂ and 1.064±0.020 for Mg(HCOO)₂·2H₂O. Despite that compounds investigated in the present work are important components for tropospheric aerosols, in general they have not been included in widely used aerosol thermodynamic models, such as E-

AIM (Clegg et al., 1998) and ISORROPIA II (Fountoukis and Nenes, 2007). The systematical and
comprehensive datasets which we have obtained in this work are highly valuable and can be used
to validate thermodynamic models if they are extended to include these compounds.
In addition, hygroscopic growth of aerosol particles was measured at room temperature for
these eight compounds. Being different from solid samples for which the onset of deliquescence
was evident, aerosol particles were found to grow in a continuous manner since very low RH (as
low as 10%), implying that these dry aerosol particles generated from aqueous droplets were
amorphous. Hygroscopic growth factors of aerosol particles at 90% RH were determined to be
1.79±0.03 and 1.67±0.03 for $Ca(NO_3)_2$ and $Mg(NO_3)_2$, 1.71±0.03 for both $CaCl_2$ and $MgCl_2$,
1.54±0.02 and 1.69±0.03 for $Ca(HCOO)_2$ and $Mg(HCOO)_2$, and 1.26±0.04 and 1.53±0.01 for
$Ca(HCOO)_2$ and $Mg(HCOO)_2$. GF at 90% show significant variation (from ~1.26 to ~1.79) for the
Ca-containing salts investigated here; among them nitrate and chloride have very similar GF
(1.79±0.03 versus 1.71±0.03), which are larger than that of formate (1.54±0.02), while acetate has
the smallest GF (1.26±0.04). Interestingly, for the four Mg-containing salts considered in this work,
the variation in GF at 90 % RH was found to be much smaller (from ~1.53 to ~1.71).
GF at 90% RH were used to derive the single hygroscopicity parameters ($\kappa$), which were
determined to be 0.49-0.56 and 0.38-0.43 for $Ca(NO_3)_2$ and $Mg(NO_3)_2$, 0.42-0.47 for both $CaCl_2$
and $MgCl_2$, 0.28-0.31 and 0.40-0.45 for $Ca(HCOO)_2$ and $Mg(HCOO)_2$, and 0.09-0.13 and 0.28-
0.29 for $Ca(HCOO)_2$ and $Mg(HCOO)_2$ aerosols, respectively. $Ca(NO_3)_2$ and $CaCl_2$ are very soluble
in water, and thus their $\kappa$ values derived from our H-TDMA experiments are consistent with those
reported by previous CCN activity measurements (Sullivan et al., 2009; Tang et al., 2015); on the
other hand, due to limited water solubilities, for $Ca(HCOO)_2$ and $Ca(CH_3COO)_2$, $\kappa$ values derived
from our H-TDMA experiments are significantly smaller than those derived from CCN activities
(Tang et al., 2015). Overall, the present work would significantly improve our knowledge in the
hygroscopic properties of Ca- and Mg-containing salts, and thereby help better understand the
physicochemical properties of mineral dust and sea salt aerosols.

## Author contribution

Mingjin Tang designed the research; Liya Guo, Peng Chao, Taomou Zong, Qinhao Lin and
Guohua Zhang did the H-TDMA experiments and analyzed the results with the assistance and
supervision of Weigang Wang, Zhijun Wu, Maofa Ge, Min Hu and Xinhui Bi; Wenjun Gu and
Yujing Tang did the VSA experiments and analyzed the data with the supervision of Yong Jie Li,
Xinming Wang and Mingjin Tang; Yong Jie Li and Mingjin Tang wrote the manuscript with the
contribution from all the other co-authors.

## Acknowledgement

This work was funded by the National Natural Science Foundation of China (91744204,
91644106 and 41675120), the Chinese Academy of Sciences international collaborative project
(132744KYSB20160036) and the special fund of State Key Joint Laboratory of Environment
Simulation and Pollution Control (17K02ESPCP). Mingjin Tang also would like to thank the CAS
Pioneer Hundred Talents program for providing a starting grant. Yujing Tang contributed to this
work as an undergraduate intern at Guangzhou Institute of Geochemistry. This is contribution No.
IS-XXX from GIGCAS.

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
