# Peer review of "A comprehensive study of hygroscopic properties of calcium- and magnesium-containing salts: implication for hygroscopicity of mineral dust and sea salt aerosols"

_Atmospheric Chemistry and Physics, 2018_

## Referee Comment (RC1) · Anonymous Referee #2 · 9 Jul 2018

This paper uses two complementary techniques to explore the water uptake properties of commonly found Mg and Ca salts in mineral dust and sea salts that are relevant to atmospherically aged particles. The manuscript is thorough and very well written. I recommend this work for publication after minor revisions.

Major Comments: I have several major comments:

1. More discussion of salts found in freshly emitted and heterogeneously processed sea salt aerosols would have balanced out the intro and discussion in this paper.

2. The conclusions and implications section would have benefited from discussion of the implications for water uptake and CCN activation of freshly emitted and processed dusts and sea salts.

3. The authors should create another table or a $\kappa$ plot with their $\kappa$GF values and to report these values in the abstract. This will attract more attention to their work.

Specific Comments: Abstract

1. I recommend pointing out that your results also fit theoretical predictions from the Clausius-Clapeyron equation and to report $\kappa$GF values obtained from this work. This is important for incorporating your results into models.

Introduction

1. Please also reference [Gaston et al., 2017] which explored the CCN activity of playa dusts.

2. Lines 75-78: CaCl2 would also be important for sea spray aerosol.

3. Lines 80-83: were these previous studies incomplete that the work warrants further investigation? How so?

Methods

1. Were diameters corrected for shape factors particularly for dry particle diameters?

Results

1. Line 414: Gaston et al., 2017 also measured $\kappa$CCN for CaCl2 and for a MgCl2 hydrate.

2. The authors are encouraged to create another table or a $\kappa$ plot with their $\kappa$GF values and to report these values in the abstract. This will attract more attention to their work.

Conclusions:

1. The authors are encouraged to point out the broader implications of their work for the water uptake and cloud nucleating properties of fresh and processed dusts and sea salts.

2. The authors are encouraged to also point out the ability of the Clausius-Clapeyron equation to predict the temperature-dependent behavior of the water uptake properties of some of the salts.

References: Gaston, C. J., K. A. Pratt, K. J. Suski, N. W. May, T. E. Gill, and K. A. Prather (2017), Laboratory studies of the cloud droplet activation properties and corresponding chemistry of saline playa dust, Environmental Science & Technology, 51(3), 1348-1356.
* * *

---

## Referee Comment (RC2) · Anonymous Referee #3 · 13 Aug 2018

**General comments:**

The authors presented detailed hygroscopic properties of Ca- and Mg-containing salts by performing both diameter and mass growth measurements using advanced instruments such as HTDMA and vapor sorption analyzer (VSA). The temperature dependences of DRH and water-to-solute ratios for three specific Ca- and Mg-containing salts were also discussed based on the corresponding VSA measurements. The techniques used in this study are valid, and the obtained data sets can be served as a database for hygroscopic properties of Ca- and Mg-containing mineral dust and sea salt particles. However, more in-depth discussion and major revisions are needed. I would recommend this manuscript to be accepted after the following comments are well addressed.

**Major comments:**

1. It is good to see some comparison results between this work and previous studies, such as CCN measurements for the same types of Ca- / Mg-containing salts. It should be noted that hygroscopic measurements in this study were mainly performed under sub-saturated conditions, while previous CCN measurements were conducted under supersaturated conditions. In this sense, how should readers understand all the comparison results of the hygroscopicity parameter, $\kappa$, between calculated in this study and derived from previous CCN measurements? How do they differ from each other, and are they really comparable? These concerns should be addressed more clearly.

   At the same time, how is the comparison between the two types of hygroscopic growth results obtained within this study, since the authors have conducted both diameter growth and mass change measurements for the same Ca- and Mgcontaining salts? How will the particle morphology or crystalline state influence the agreement between these two types of water uptake measurements? Further discussion is needed to clarify the abovementioned points.

2. In the Experimental section, the authors have mentioned that the H-TDMA system was routinely checked with 100 nm $(NH_4)_2SO_4$ and NaCl particles. How were the H-TDMA calibration results and how did them compare with theoretical values, since the absolute uncertainty in measured RH was stated to be ±2% (e.g., in Table 4)? It will be good to show some calibration results to verify the reliability of data obtained from the H-TDMA measurements. In addition, are the GF results reported in this work after data inversion, as no further detailed information has been mentioned when displaying Eq. (3) in Sect. 3.1.4? What kinds of inversion algorithms were applied to the H-TDMA data? These need to be explicitly provided and well referenced.

   Another issue is about the $\kappa_{gf}$ results calculated from the H-TDMA measurements at 12 different RH conditions. How is the variability in derived $\kappa_{gf}$ results for a specific salt, as the authors have suggested that only the corresponding $\kappa_{gf}$ results at 90% RH were used for comparison with previous CCN studies?

3. How will the obtained hygroscopic data be compiled into the thermodynamic models? How to consider the crystalline reference state for those Ca- / Mg-containing sea salt or dust particles if no detailed information was available?

**Specific comments:**

1. **Abstract**, Page 2, line 38

   *"All the aerosol particles studied in this work, very likely to be amorphous, started to grow at very low RH …"*

The "amorphous" statement appeared here and elsewhere in the manuscript. How was this amorphous state determined? How was it identified from the possible supersaturated droplet condition?

2. **Introduction**, Page 3, line 56

*"Mineral dust aerosol has an average lifetime of a few days in the atmosphere and can thus be transported over thousands of kilometers."*

The "a few days" here is a bit confusing, as it would be inappropriate to use "a few days" if longer than weeks. How was a-few-days average lifetime estimated from the transport distance of over thousands of kilometers, and what was the average wind velocity during long-range transportation?

3. **Experimental section**, Page 6, line 132

How is the stability of RH during the H-TDMA measurements? Are the comparisons with previous hygroscopic studies (e.g., in Sect. 3.1.4) always for the same dry diameter? Is any size dependence of the measured hygroscopic properties observed in this work?

Page 7, line 150: *"...after that, RH was set to 0% to dry the sample again."*

How was RH = 0% achieved and defined in this study? Could it be really 0, and is 0% RH appropriate for the real experimental conditions?

Page 7, line 153: *"... until a significant increase in sample mass was observed..."*

How did the authors define the "significant increase" in this study? Accordingly, can you provide any specific details for the threshold value of mass change?

4. **Results and discussion**, Page 8, line 164

*"An abrupt and significant increase in sample mass was observed when RH was increased from 52 to 53%, suggesting that the deliquescence occurred between 52*

*and 53% RH."*

Did the mass change here really suggest "deliquescence" or likely due to the mass increase by surface water adsorption during hydration process? This needs to be explained in the manuscript.

5. Page 11, line 225

*"... $\Delta H_s$ is the enthalpy of solution (J mol$^{-1}$)"*

Shouldn't $\Delta Hs$ reflect the change of enthalpy?

6. Page 12, line 238

*"... the estimated $\Delta Hs$ value for $MgCl_2·6H_2O$ had a large uncertainty (probably a factor of >2) due to the very small dependence of its DRH on temperature."*

How was the "*probably a factor of >2*" estimated? Any data results can be shown to verify this statement?

7. Page 12, line 244

*"The mass change, relative to that at 0% RH, can be used to calculate water-to-solute ratios (WSR, ...)"*

It is important to demonstrate that the measurements were conducted when particles have reached an equilibrium state under completely dry conditions before calculating the "water-to-solute ratios". Accordingly, are the $m/m_0$ values (Table 1) at RH < DRH conditions due to hygroscopic growth or surface water adsorption during hydration?

8. Page 22, Eq. 4

What kinds of assumptions or simplification have been made to obtain this equation? How to understand the influences of Solute effect and Kelvin effect, since no

explicitly relevant parameters were displayed in the current format?

9. Page 23, line 419

   "*To our knowledge, CCN activities of Mg(NO₃)₂ and MgCl₂ aerosols have not been experimentally explored yet, and $\kappa_{ccn}$ were predicted to be 0.8 for Mg(NO₃)₂, 0.3 for Mg(NO₃)₂·6H₂O, and ~1 for MgCl₂ ...*"

   How were the $\kappa_{ccn}$ values for $Mg(NO_3)_2$ and $MgCl_2$ estimated here? What kinds of assumptions were applied into the corresponding $\kappa$ calculation?

10. Page 24, line 449

    "*... the DRH value of Ca(CH₃COO)₂·H₂O, measured by further experiments, was determined to be 90.5±1.0 % at 25 ºC in our work.*"

    Where can readers find the corresponding details for the "*further experiments*" used in the estimation of the DRH value here? Necessary information is needed.

11. Page 29, line 542

    "*... This is largely caused by the difference in water solubilities of Ca(NO₃)₂, CaCl₂, Ca(HCOO)₂ and Ca(CH₃COO)₂.*"

    Also in **Conclusion section**, Page 32, line 601

    "*Ca(NO₃)₂ and CaCl₂ are very soluble in water, and thus their $\kappa$ values derived from our H-TDMA experiments are consistent with those reported by previous CCN activity measurements (Sullivan et al., 2009; Tang et al., 2015); on the other hand, due to limited water solubilities, for Ca(HCOO)₂ and Ca(CH₃COO)₂, $\kappa$ values derived from our H-TDMA experiments are significantly smaller than those derived from CCN activities measured in a previous study (Tang et al., 2015).*"

    The authors have attributed the discrepancy between $\kappa_{gf}$ and $\kappa_{ccn}$ mainly to the difference in water solubility of these Ca-containing salts, however, without

discussing any possible differences in the particle water uptake measurements under both sub- and supersaturated conditions. In addition to this main concern raised in **Major comments #1**, how did the authors evaluate the uncertainties in calculated $\kappa_{gf}$ results from H-TDMA measurements in this study, e.g., uncertainties related to GF observation, RH fluctuation, and $\kappa_{gf}$ derivation?

12. **Table** 1

What does it actually refer to when stating "*All the errors (±1 σ) are statistical only*" in all the Tables? Are they standard deviations?

13. **Table** 2

The authors have declared that "*WSR were only calculated for RH exceeding the DRH*". According to the DRH value of $Ca(NO_3)_2 \cdot 4H_2O$ at 5 °C presented in Table 1, i.e., 60.5 ± 1.0 %, does it mean that the corresponding WSR results at 50% and 60% RH in Table 2 are inaccurate?

14. **Tables** 4 and 6

GF results measured with the H-TDMA setup at different RH conditions were presented in these tables. However, the corresponding $\kappa_{gf}$ results, if also shown in the tables or displayed in separate plots, would be more straightforward for readers when in comparison with previous hygroscopic results.

15. **Figures** 1, 3, and 5

Are the RH values shown in the y-axis corresponding to the specific RH set points or real RH conditions achieved during the experiments? How to explain the decrease in normalized mass during 700 ~ 1,000 min in Figure 1?

How did the authors define when the particles were completely dry and the particle mass reached the lowest value, as which was applied in the normalization of particle mass with changing RH conditions? For example in Figure 3, particles didn't seem

to be completely dried at ~ 4,200 min when the normalized mass was taken as 1.0.

According to the x-axis in these figures, the time scales corresponding to a specific experiment are significantly different. Is the time taken for each experiment of the eight salts always so different? Have the authors tried to repeat these VSA experiments, and how were the replicability and corresponding uncertainties in these measurements?

16. **Figures** 6

How to explain the decrease trend in observed GF (i.e., GF < 1.0 at around 60~80% RH conditions) in Figure 6b? More detailed discussion should be provided in the corresponding data interpretation sections.

**Technical corrections:**

1. **Abstract**, Page 2, line 40

   *"... were found to range from 1.26±0.04 for Ca(HCOO$_2$)$_2$ and 1.79±0.03 for Ca(NO$_3$)$_2$ ..."*

2. **Introduction**, Page 4, line 72

   *"... CaCl$_2$ from heterogeneous reactions with nitrogen oxides (Goodman et al., 2000; Liu et al., 2008a; Li et al., 2010; Tang et al., 2012; Tan et al., 2016) and HCl (Santschi and Rossi, 2006)"*

   It seems that different expression formats were randomly used for the chemical species mentioned in this manuscript. Similar issues can be found elsewhere.

3. **Sect. 3.1.4**, Page 20, line 365

   *"In the second study (Jing et al., 2018), GF were determined to be 1.56 and 1.89 at*

*80 and 90% RH."*

Page 23, line 414

*"2) For CaCl$_2$ aerosol,  $\kappa_{ccn}$ were measured to be 0.46-0.58…"*

4. **Sect. 3.2**, Page 24, line 436

*"The average ratio of sample mass at 95% RH to that at 0% RH was determined to be  1.043 ± 0.018 for Ca(HCOO)$_2$ and 1.028 ± 0.008 for Mg(HCOO)$_2$·2H$_2$O (not shown in Figure 5), probably indicating that the DRH values were >95% for both compounds at 25 °C."*

5. Page 25, line 462

*"The RH over the saturated Mg(CH$_3$COO$_2$)$_2$ solution at ~23 $^{o}C$ was measured …"*

Please check all the units carefully throughout the manuscript.

6. Page 26, line 480

*"Table 5 also reveals that a small increase in sample mass (by ~3%, relative to that at 0% RH) was observed for Mg(CH$_3$COO)$_2$·4H$_2$O when RH was increased to 70% before its deliquescence occurred."*

It is better to clarify "its" here.

7. **Conclusion section**, Page 30, line 560

*"In this work, phase transition and hygroscopic growth of these eight compounds were systematically examined …"*

The "these eight" here might be unclear to readers if without specific introduction in advance.

---

## Referee Comment (RC3) · Anonymous Referee #1 · 17 Aug 2018

The authors presented a comprehensive study of the hygroscopic properties of calcium- and magnesium- containing salts using a vapor sorption analyzer and a HTDMA. The change of sample mass with RH and the corresponding DRH value was reported for these eight compounds together with their hygroscopic growth factor values at 90% RH. The dataset is rich, however, the comparison and discussion is not sound that major revisions are needed. The manuscript may be acceptable for publication in ACP after the following concerns are fully addressed.

Major comments:

1. This work used a vapor sorption analyzer to measure the change of sample mass with relative humidity and the deliquescence relative humidity of eight different compounds. However, as I understand, the materials the author used in this study are not atmospheric particles, but actually bulk samples. Please clarify how these results represent atmospheric conditions. The hygroscopic properties and DRH of aerosol particles would probably deviate significantly from that of bulk samples. Please extend your manuscript with explicit discussions regarding these issues to prove the significance for atmospheric research.

2. What is the relation between the mass growth factor and mobility growth factor as measured by two independent methods? Extensive works have been performed to measure the hygroscopic growth factor of atmospheric relevant compounds from previous studies. How to compare the mass growth factor obtained by the vapor sorption analyzer with their results and what kind of uncertainties should be taken into account?

3. The DRH values of these studied compounds can also be measured by your HTDMA setup. Why the results obtained by your HTDMA did not agree with the ones from the VSP. What is the explanation for the discrepancies? Since the VSP measures the bulk samples, are these results obtained from the VSP measurements applicable in atmospheric research. Could you also please plot the GF-PDF for each compounds measured by the HTDMA? Are there unimode or bimode for your growth factor distributions at different RHs?

4. For the inorganic species you studied, you stated they are important components in mineral dust or sea salt particles. However, for the VSA measurements, you studied their hygroscopic properties of their hydrate forms, while you measured the HGF of these compounds in their anhydrous

state using HTDMA. As I see, the hygroscopic properties of these compounds vary significantly between their anhydrous states and hydrate states (for instance, line 420-421). I feel difficult to relate your results with your introduction and objectives. Which state exist in the real atmosphere? Moreover, which state is hygroscopic and which values should we use for further study? Please clarify and be consistent through your whole manuscript. Otherwise, give your explanations.

5. For your conclusion part, it is more like a summary of your results without any atmospheric implications. Please rephrase it and highlight its atmospheric applications.

Specific comments:

1. Line 93-97: These two statements are in conflict with each other.

2. Line 128-131: How long is your humidifier and what is the flow rate? And what is the accuracy of your RH measurements, please give its uncertainty.

3. Line 139: What do you mean by a particle sample? Did you generate particles and measure the mass of these particles? If not, please rephrase it.

4. Line 210-211: What is the possible reason for the deviations?

5. Line 222-223: Give proper reference for Eq. 1 in your manuscript, the original source but not only these who also cites it. In addition, which solubility (at which temperature condition) you used for your calculation, as it also depends on temperature.

6. Line 225: Enthalpy of what? Deliquensece or dissolution?

7. Line 246: Could you give proper explanation why WSR increase with a decreasing in temperature.

8. Line 253, Table 2: How could you get the WSR value for Ca(NO3)2 at 50% RH, as it did not deliquesce yet according to your previous results in Table 1 (DRH as 60.5%).

9. Line 258-261: I don't understand the sentences.

10. Line 265: What concentration do you mean here? Bulk solution or droplet? Please be specific.

11. Line 286: Did you also observe similar phenomena for the other two inorganic compounds for phase transition, as it seems to be according to Fig. 1 in your manuscript.

12. Line 311-312: I don't think this is fairly new result as it is still bulk sample. We should always consider size effect as it is atmospheric or at least particle-relevant.

13. Line 337: What is the stuff after atomizing? Are they in hydrate state or not?

14. Line 363-365 and line 379-382: So should we use the dry diameter selected by the DMA (100nm) or not. If yes, it seems your results did not agree with the ones from Gibson et al., (2006) in line 363-365, but agreed in line 379-382. Please clarify.

15. Extra cautions must be taken by introducing several scientific terms in the manuscript. For instance, in line 221-223. What is the scientific reason to study the temperature-dependence of DRH and its enthalpy value? Please clarify. For instance, in line 258-263, why you studied the water-solute ratio and what is this variable used for? What is the relation between water activity and water-solute ratio? And how you converted it to each other in details?

16. Which particle size did you selected during the HTDMA measurements? In Eq. 4 in your manuscript, where is the Kelvin term? Please use the correct formulation and make further comparison. For instance, in line 410-411, which particle size or which supersaturation they selected in their CCN measurements?

17. In addition, please rephrase your discussion part and make sound comparisons with the other studies. For instance, line 367-368, GF of Ca(NO3)2 aerosols was measured to be 1.79 in your work, while Jing et al., (2018) reported it to be 1.89 at 90%RH. In line 387-389, Park et al., (2009) measured the GF of CaCl2 to be 1.59 at 90% RH and the measured value from your result was 1.71. There were some differences (around 7%) but not always in good agreement as you stated in the manuscript between your results and the ones from others. Please give proper discussions.

---

## Author Comment (AC1) · 23 Sep 2018

Comments by Referees are in blue.

Our replies are in black.

Changes to the manuscript are highlighted in red both in here and in the revised manuscript.

**Reply to Ref #3**

**General comments:**

The authors presented detailed hygroscopic properties of Ca- and Mg-containing salts by performing both diameter and mass growth measurements using advanced instruments such as HTDMA and vapor sorption analyzer (VSA). The temperature dependences of DRH and water-to-solute rations for three specific Ca- and Mg- containing salts were also discussed based on the corresponding VSA measurements. The techniques used in this study are valid, and the obtained date sets can be served as a database for hygroscopic properties of Ca- and Mg-containing mineral dust and sea salt particles. However, more in-depth discussion and major revisions are needed. I would recommend this manuscript to be accepted after the following comments are well addressed.

**Reply:** We would like to thank ref #3 for his/her very positive evaluation on our manuscript and detailed comments which have significantly helped us improve the manuscript. We have addressed all the comments adequately, as detailed blow.

**Major comments:**

1. It is good to see some comparison results between this work and previous studies, such as CCN measurements for the same types of Ca- / Mg-containing salts. It should be noted that hygroscopic measurements in this study were mainly performed under sub-saturated conditions, while previous CCN measurements were conducted under supersaturated conditions. In this sense, how should readers understand all the compassion results of the hygroscopicity parameter, $\kappa$, between calculated in this study and derived from previous CCN measurements? How do they differ from each other, and are they really comparable? These concerns should be addressed more clearly.

**Reply:** Ideally aerosol-water interaction under both subsaturation and supersaturation can be described by a constant single hygroscopicity parameter; nevertheless, discrepancies have been widely reported, due to 1) solution ideality; 2) solubility limit; 3) surface tension. In the revised manuscript we have added a few sentences (line 458-468) to briefly discuss the comparability between $\kappa_{gf}$ and $\kappa_{ccn}$. By doing so and referring readers to a few key references, we have provided necessary theoretical background to understand these comparisons.

At the same time, how is the comparison between the two types of hygroscopic growth results obtained within this study, since the authors have conducted both diameter growth and mass change measurement for the same Ca- and Mg-containing salts? How will the particle morphology or crystalline state influence the agreement between these two types of water uptake measurement? Further discussion is needed to clarify the abovementioned points.

**Reply:** This is a very good comment, and in the revised manuscript (line 635-665) we have added one section, entitled "Comparison between H-TDMA and VSA measurements", to compare the two types of results. While details can be found in our revised manuscript, here we outline our major points in brief:

1) When RH are higher than the DRH, both bulk samples used in VSA measurements and aerosol particles used in H-TDMA experiments would deliquesce to form aqueous solutions, and measured mass change and diameter change can be linked by solution densities which also depend on water activity (i.e. RH).

2) When RH are lower than DRH, the two types of results cannot be reconciled, since VSA measured the hygroscopic properties of crystalline samples while H-TDMA measured the hygroscopic growth of amorphous aerosol particles.

2. In the Experiment section, the authors have mentioned that the H-TDMA system was routinely checked with 100 nm $(NH_4)_2SO_4$ and NaCl particles. How were the H-TDMA calibration results and how did them compare with theoretical values, since the absolute uncertainty in measured RH was state to be ±2% (e.g., in Table 4)? It will be good to show some calibration results to verify the reliability of data obtained from the H-TDMA measurement. In addition, are the GF results reported in this work after data inversion, as no further detailed information has been mentioned when displaying Eq. (3) in Sect.3.1.4? What kinds of inversion algorithms were applied to the H-TDMA data? These need to be explicitly provided and well referenced.

**Reply:** The following changes have been made in the revised manuscript:

1) The agreement between our measurements and theoretical predictions is very good. Since technical descriptions of our H-TDMA, including its experimental validation, have been detailed in our previous studies, we choose to refer interested readers to our previous studies for further information, and we have rephrased the relevant sentence in the revised manuscript (line 454-158) to provide necessary information: "The performance of the H-TDMA setup was routinely checked by measuring the hygroscopic growth of 100 nm $(NH_4)_2SO_4$ and NaCl aerosol particles, and good

agreement between measured hygroscopic growth curves with those predicted using the E-AIM model (Clegg et al., 1998) was always found for these two types of aerosol particles, as detailed in our previous studies (Jing et al., 2016; Peng et al., 2016)."

2) The TDMAinv algorithm (Gysel et al., 2009) was applied to the H-TDMA data. In the revised manuscript (line 149-150) we have included one sentence to clarify it: "The TDMAinv algorithm (Gysel et al., 2009) was applied to the H-TDMA data."

Another issue is about the $\kappa_{gf}$ results calculated from the H-TDMA measurements at 12 different RH conditions. How is the variability in derived $\kappa_{gf}$ results for a specific salt, as the authors have suggested that only the corresponding $\kappa_{gf}$ results at 90% RH were used for comparison with previous CCN studies?

**Reply:** CCN measurements are carried out at supersaturation when aerosol particles (or cloud droplets) are highly diluted droplets. H-TDMA measurements are carried out at subsaturation when aerosol particles are much more concentrated droplets, and these droplets become more diluted at higher RH. Therefore, when one wants to reconcile H-TDMA measurements with CCN measurements, growth factors measured at high RH are always used. This is why GF measured at 90% RH, instead of those at lower RH, have been used in our and many previous studies to calculate $\kappa_{gf}$, as we stated in our original manuscript.

3. How will the obtained hygroscopic data be compiled into the thermodynamic models? How to consider the crystalline reference state for those Ca- / Mg-containing sea salt or dust particles if no detailed information was available?

**Reply:** In the revised manuscript, we have added a few sentences (line 666-700) to discuss how our data can be used by aerosol thermodynamic models. In brief, key outputs of common aerosol thermodynamic models include RH-dependent water-to-solute ratios and volumes of solutions for RH above DRH; since both parameters were measured as a function of RH in our work, experimental data obtained in our work can be used to assess the performance of aerosol thermodynamic models. More details can be found in our revised manuscript.

For RH below DRH, indeed no detailed information is available yet regarding the crystalline state of aerosol particles investigated in our work. Since H-TDMA measurements are of direct atmospheric relevance, we suggest that for RH below DRH, H-TDMA results should be used for atmospheric applications; in addition, we have discussed which types of experiments can be used to reveal the crystalline states of aerosol particles examined in our work.

1. Abstract, Page 2, line 38: The "amorphous" statement appeared here and elsewhere in the manuscript. How was this amorphous state determined? How was it identified from the possible supersaturated droplet condition?

**Reply:** Based on our observation that aerosol particles showed continuous water uptake, we concluded that these particles were likely to be amorphous. Our conclusion is also supported by a number of previous studies using vibrational spectroscopy, EDB and H-TDMA. This has been also discussed in our original manuscript (line 343-350).

2. Introduction, Page 3, line 56: The "a few days" here is a bit confusing, as it would be inappropriate to use "a few days" if longer than weeks. How was a-few-days average lifetime estimated from the transport distance of over thousands of kilometers, and what was the average wind velocity during long-range transportation?

**Reply:** To make the statement more specific, in the revised manuscript (line 57) the sentence has been changed to "Mineral dust aerosol has an average lifetime of 2-7 day in the atmosphere and can thus be transported over thousands of kilometers (Textor et al., 2006; Uno et al., 2009)."

3. Experimental section, Page 6, line 132: How is the stability of RH during the H-TDMA measurements?

**Reply:** We have included one sentence in the revised manuscript (line 153-154) to make the uncertainties in RH clear: "The absolute uncertainties in RH were estimated to be within ±2%." The stated uncertainties here took into account the RH stability in each individual experiments as well as RH reproducibility in replicate experiments.

Are the comparisons with previous hygroscopic studies (e. g., in Sect. 3.1.4) always for the same dry diameter? Is any size dependence of the measured hygroscopic properties observed in this work?

**Reply:** Except for Park et al. (2009), all the other previous studies used a dry particles diameter of 100 nm. In the revised manuscript we have provided the size information.

Park et al. (2009) explored the hygroscopic growth of $CaCl_2$ and $MgCl_2$ aerosol particles at three different diameters (20, 30 and 50 nm), and no size effect was observed. In the revised manuscript we have added one sentence (line 426-428) to provide more information: "Three dry diameters (20, 30 and 50 nm) were used for $CaCl_2$ and $MgCl_2$ aerosol particles (Park et al., 2009), and no significant size dependence of their hygroscopic properties was observed."

Page 7, line 150: How was RH=0% achieved and defined in this study? Could it be really 0, and is 0% RH appropriate for the real experimental conditions?

**Reply:** The actual RH was measured to be <1%. In the revised manuscript (line 171) we have stated it more clearly: "after that, RH was set to 0% (the actual RH was measured to be <1%) to dry the sample again."

Page 7, line 153: How did the authors define the "significant increase" in this study? Accordingly, can you provide any specific details for the threshold value of mass change?

**Reply:** In our work, a significant increase in mass was considered to have occurred if the observed increase was larger than the magnitude of baseline drift. To make it clearer, in the revised manuscript (line 175) we have changed the sentence to "RH was then increased stepwise with an increment of 1% until a significant increase in sample mass (when compared to the baseline drift) was observed, and the RH at which the sample mass showed a significant increase was equal to its DRH."

4. Results and discussion, Page 8, line 164: Did the mass change here really suggest "deliquescence" or likely due to the mass increase by surface water adsorption during hydration process? This needs to be explained in the manuscript.

**Reply:** In the revised manuscript, at the end of this paragraph (line 191-193) we have added one sentence to further justify our claim: "Therefore, its DRH was measured to be 52.5±0.5 %. It should be noted that the mass change was >15% when RH was changed from 52 to 53%, as shown in Figure 1a; such a large mass increase cannot be caused by water adsorption."

5. Page 11, line 225: Shouldn't ΔHs reflect the change of enthalpy?

**Reply:** I checked the textbook by Seinfeld and Pandis (2006), and ΔHs is termed as "enthalpy of dissolution" instead of "enthalpy of solution". We have corrected it in the revised manuscript (line 258).

6. Page 12, line 238: How was the "probably a factor of >2" estimated? Any data results can be shown to verify this statement?

**Reply:** We have carefully considered this comment. Because the change in DRH with temperature may be insignificant, in the revised manuscript we have chosen not to report $\Delta H_s$ for $MgCl_2 \cdot 6H_2O$ because such estimation may have very large errors. In the revised manuscript, we have revised this sentence (line 266-268): "The variation of DRH values with temperature (5-30 ℃) was very small and even insignificant for $MgCl_2 \cdot 6H_2O$; as a result, we did not attempt to

estimate the $\Delta H_s$ values for $MgCl_2 \cdot 6H_2O$ using Eq. (2) since such estimation would have large errors." In addition, we have updated Table 1 accordingly.

7. Page 12, line 244: It is important to demonstrate that the measurements were conducted when particles have reached an equilibrium state under completely dry conditions before calculating the "water-to-solute ratios". Accordingly, are the m/m0 values (Table 1) at RH < DRH conditions due to hygroscopic growth or surface water adsorption during hydration?

**Reply:** Yes, in our work we used the mass of the sample in equilibrium to calculate water-to-solute ratios. This can be confirmed in the following two aspects: 1) in order to ensure that the equilibrium was reached, in our measurement we only changed RH to the next value when the sample mass change was <0.1% within 30 min, as explained in Section 2.2; 2) our measured water-to-solute ratios agree very well with those reported by previous work (for example, as shown in Table 2), further suggesting that equilibria were reached.

Changes in $m/m_0$ for RH below DRH were due to water adsorption/desorption and baseline drift, and in the revised manuscript we have added one sentence (line 275-278) to further explain it: "Small changes in $m/m_0$ (typically <2%) were observed for some compounds (as shown in Tables 2 and 6) when RH was below corresponding DRH values, mainly due to water adsorption/desorption and baseline drift."

8. Page 22, Eq. 4: What kinds of assumptions or simplification have been made to obtain this equation? How to understand the influences of Solute effect and Kelvin effect, since no explicitly relevant parameters were displayed in the current format?

**Reply:** The Kelvin effect is negligible for hygroscopic growth of aerosol particles with a dry diameter of 100 nm. In the revised manuscript (line 443-454) we have provided the original equation which takes into account the Kelvin effect and explained how Eq. (4) is derived. Please refer to the revised manuscript for details.

9. Page 23, line 419: How were the $\kappa_{ccn}$ values for $Mg(NO_3)_2$ and $MgCl_2$ estimated here? What kinds of assumptions were applied into the corresponding $\kappa$ calculation?

**Reply:** The main assumption is solution ideality. In the revised manuscript we have added one sentence (line 493-495) to clarify it: "These calculations were performed using the Köhler theory, assuming solution ideality (Kelly et al., 2007)."

10. Page 24, line 449: Where can readers find the corresponding details for the "further experiments" used in the estimation of the DRH value here? Necessary information is needed.

**Reply:** As suggested, in the revised manuscript we have rephrased this sentence (line 519-521) to provide necessary details: "In further experiments significant increase in sample mass was observed when RH was increased from 90 to 91% for $Ca(CH_3COO)_2 \cdot H_2O$ at 25 °C, suggesting that its DRH was measured to be 90.5±1.0 %."

11. Page 29, line 542; also in Conclusion section, Page 32, line 601: The authors have attributed the discrepancy between $\kappa_{gf}$ and $\kappa_{ccn}$ mainly to the difference in water solubility of these Ca-containing salts, however, without discussing any possible differences in the particle water uptake measurements under both sub- and supersaturated conditions.

**Reply:** Since this concern was also raised in Major comments #1, it has been addressed in our reply to Major comments #1. To summarize, in the revised manuscript we have added a few sentences (line 458-468) to provide necessary theoretical background to understand the comparison between $\kappa_{gf}$ and $\kappa_{ccn}$.

In addition to this main concern raised in Major comments #1, how did the authors evaluate the uncertainties in calculated kgf results from H-TDMA measurements in this study, e.g., uncertainties related to GF observation, RH fluctuation, and $\kappa_{gf}$ derivation?

**Reply:** We have taken into account the uncertainties in measured GF in calculating $\kappa_{gf}$, and in the revised manuscript (line 472-474) we have added one sentence to clarify it:"The uncertainties in our derived $\kappa_{gf}$ have taken into account the uncertainties in measured GF at 90% RH."

12. Table 1: What does it actually refer to when stating "All the errors (± 1 σ) are statistical only" in all the Tables? Are they standard deviations?

**Reply:** Yes, they are standard deviations. In the revised manuscript, we have change the table captions to make this clearer.

13. Table 2: The authors have declared that "WSR were only calculated for RH exceeding the DRH". According to the DRH value of $Ca(NO_3)_2 \cdot 4H_2O$ at 5 °C presented in Table 1, i.e., 60.5 ± 1.0%, does it mean that the corresponding WSR results at 50% and 60% RH in Table 2 are inaccurate?

**Reply:** The referee is corrected, and WSR could only be calculated for $Ca(NO_3)_2 \cdot 4H_2O$ at 5 °C when RH was >60%. We made an error when we prepared this table for the original manuscript, and in the revised manuscript we have corrected this error.

14. Table 4 and 6: GF results measured with the H-TDMA setup at different RH conditions were presented in these tables. However, the corresponding $\kappa_{gf}$ results, if also shown in the tables or displayed in separate plots, would be more straightforward for readers when in comparison with previous hygroscopic results.

**Reply:** This is a very good point. As suggested, we have included a new table (Table 5) in our revised manuscript to compare our measured $\kappa_{gf}$ with $\kappa_{ccn}$ measured in previous work. We have also changed relevant discussion accordingly (please refer to Sections 3.1.4 and 3.2.2 for further details).

15. Figures 1, 3, and 5: Are the RH values shown in the y-axis corresponding to the specific RH set points or real RH conditions achieved during the experiments?

How to explain the decrease in normalized mass during 700 ~ 1,000 min in Figure 1?

How did the authors define when the particles were completely dry and the particle mass reached the lowest value, as which was applied in the normalization of particle mass with changing RH conditions? For example in Figure 3, particles didn't seem to be completely dried at ~ 4,200 min when the normalized mass was taken as 1.0.

**Reply:** 1) Specific RH set points were plotted in these three figures, and the difference between actual and set RH was <1%, as stated in Section 2.2. It may take several minutes (estimated using the flow rate and the chamber volume) to reach the set RH when RH was changed, and this was short because at each RH the sample was in contact with its environment for at least 30 min.

2) This small decrease in sample mass was likely due to desorption of residual water. In the revised manuscript we have added one sentence (line 193-196) to explain it: "The continuous but small decrease in sample mass (about 1% in total) with time (around 500-1000 min) before deliquescence took place, as displayed in Figure 1a, was likely caused by desorption of residual water contained by the sample under investigation."

3) When plotting these three figures, we did not always to normalize the sample mass to the dry particle mass. This is why the right $y$-axis in these three figures is labelled as normalized sample mass instead of $m/m_0$. As stated in Section 2.2, the equilibrium was considered to be reached only when the sample mass change was <0.1% within 30 min, and the data shown in Figure 3 fulfilled this criterion. For further discussion on the criterion to determine if the sample was completely dry, please refer to **our reply to Specific comment #7**.

According to the x-axis in these figures, the time scales corresponding to a specific experiment are significantly different. Is the time taken for each experiment of the eight salts always so different? Have the authors tried to repeat these VSA experiments, and how were the replicability and corresponding uncertainties in these measurements?

**Reply:** The time to reach the equilibrium was largely determined by the dry sample mass (for the same compound, it took longer to reach the equilibrium if the dry sample mass was larger) and also varied with compounds. We have added one sentence (line 179-181) in the last paragraph of Section 2.2 to clarify it: "The time to reach a new equilibrium varied with compounds and was largely affected by the dry sample mass, i.e. samples with larger dry mass would took longer to reach the equilibrium."

Each VSA measurement was repeated at least three times, and the reproducibility was very good (as shown in Tables 1-2). In the last paragraph of Section 2.2, we have added one sentence (line 176-177) to clarify it: "Each measurement was repeated for at least three times, and the average value and standard deviation were reported."

16. Figures 6: How to explain the decrease trend in observed GF (i. e., GF < 1.0 at around 60~80% RH conditions) in Figure 6b? More detailed discussion should be provide in the corresponding data interpretation sections.

**Reply:** We believe that such a decrease may not be significant if the uncertainties in GF measurements were considered. At the end of this paragraph we have added two sentences (line 583-587) to discuss this issue: "Careful inspection of Figure 6b and Table 6 reveals that a small decrease in GF from 1.03±0.01 to 1.00±0.01 for $Ca(CH_3COO)_2$ aerosol when RH was increased from 50 to 70%. Since GF is typically expected to increase with RH, the small decrease in GF (~0.03) for RH between 50 and 70% may reflect the uncertainties in GF measurements (i.e. our H-TDMA measurements cannot resolve a GF difference as small as 0.03)."

Technical corrections:

**Reply:** We appreciate ref #3 very much for reading our manuscript very carefully and pointing out typos in our original manuscript. All the corrections have been implemented in our revised manuscript.

---

## Author Comment (AC2) · 23 Sep 2018

Comments by Referees are in blue.

Our replies are in black.

Changes to the manuscript are highlighted in red both in here and in the revised manuscript.

**Reply to Ref #2**

This paper uses two complementary techniques to explore the water uptake properties of commonly found Mg and Ca salts in mineral dust and sea salts that are relevant to atmospherically aged particles. The manuscript is thorough and very well written. I recommend this work for publication after minor revisions.

**Reply:** We would like to thank Ref #2 for his/her very positive review of our manuscript. These comments, which largely helped us improve our manuscript, have been adequately addressed in our revised manuscript, as detailed below.

**Major Comments:**

I have several major comments: 1. More discussion of salts found in freshly emitted and heterogeneously processed sea salt aerosols would have balanced out the intro and discussion in this paper.

**Reply:** In the revised manuscript we have added a few sentence (line 78-83) in the introduction section to further explain the relevance of our work for sea salt aerosol and saline mineral dust, as suggested.

In addition, we have added a new section (Section 3.3.2, line 666-700), entitled "Atmospheric implications", to further discuss atmospheric implications of our work.

2. The conclusions and implications section would have benefited from discussion of the implications for water uptake and CCN activation of freshly emitted and processed dusts and sea salts.

**Reply:** We fully agree with the referee. In the revised manuscript we have changed the title of Section 4 to "Summary and Conclusion", since this section is more like a summary of our work.

Furthermore, we have added a new section (Section 3.3.2, line 666-700), entitled "Atmospheric implication", to discuss the implications of our work for mineral dust, saline mineral dust and sea salt aerosols. Please refer to Section 3.3 in the revised manuscript for details.

3. The authors should create another table or a $\kappa$ plot with their $\kappa$(GF) values and to report these values in the abstract. This will attract more attention to their work.

**Reply:** The following changes have been implemented in the revised manuscript:

1) We have created a new table (Table 5, Page 26) to compare $\kappa_{gf}$ measured in this work with $\kappa_{ccn}$ measured by previous studies, and relevant text in Section 3 has been updated accordingly.

2) We have also mentioned $\kappa_{gf}$ results in the abstract (line 40-43): "The hygroscopic growth factors at 90% RH were found to range from 1.26±0.04 for $Ca(HCOO)_2$ and 1.79±0.03 for $Ca(NO_3)_2$, and the single hygroscopicity parameter ranged from 0.09-0.13 for $Ca(CH_3COO)_2$ to 0.49-0.56 for $Ca(NO_3)_2$."

**Specific Comments:**

Abstract

1. I recommend pointing out that your results also fit theoretical predictions from the Clausius-Clapeyron equation and to report $\kappa(GF)$ values obtained from this work. This is important for incorporating your results into models.

**Reply:** The following changes have been made in the revised manuscript:

1) In the abstract (line 34-35), we have included the following sentence to point out that our results fit theoretical predictions: "We further found that the dependence of DRH on temperature can be approximated by the Clausius-Clapeyron equation."

2) We have also mentioned $\kappa_{gf}$ results in the abstract (line 40-43), as suggested.

Introduction

1. Please also reference [Gaston et al., 2017] which explored the CCN activity of playa dusts.

**Reply:** In the revised manuscript (line 80-83), the work by Gaston et al. was cited to mention the role of $CaCl_2$ in saline dust particles: "Furthermore, the CCN activity of saline mineral dust was explored (Gaston et al., 2017), and good correlations were found between the CCN activities of saline mineral dust particles and the abundance of the soluble components (e.g., $CaCl_2$) they contained."

2. Lines 75-78: $CaCl_2$ would also be important for sea spray aerosol.

**Reply:** The following change has been made in the revised manuscript (line 76-80) to mention the importance of $CaCl_2$ in sea salt aerosol: "In addition, $MgCl_2$ and $CaCl_2$ are important components in sea salt (as known as sea spray) aerosol. The presence of $MgCl_2$ and $CaCl_2$, in addition to NaCl, can alter the hygroscopicity of sea salt aerosol (Gupta et al., 2015; Zieger et al., 2017); to be more specific…"

3. Lines 80-83: were these previous studies incomplete that the work warrants further investigation? How so?

**Reply:** As we pointed out in the original manuscript (80-83), hygroscopic growth of these aerosol particles was only investigated by one or two studies, and therefore further studies are warranted.

Methods

1. Were diameters corrected for shape factors particularly for dry particle diameters?

**Reply:** No shape factors were used in our work to correct the dry particle diameters. In Section 2.1 of the revised manuscript (line 141-150), we have added one paragraph to further explain the D-TMDA method. The reviewer is kindly referred to the revised manuscript for further details.

Results

1. Line 414: Gaston et al., 2017 also measured $\kappa$(CCN) for $CaCl_2$ and for a $MgCl_2$ hydrate.

**Reply:** We would like to thank the ref #2 for bringing our attention to the results reported by Gaston et al. (2017). We were aware of this paper, but did not pay attention to the supplementary information in which $\kappa_{ccn}$ was reported for $CaCl_2$ and $MgCl_2$. The following changes have been made in the revised manuscript:

1) We have created a table (Table 5, Page 26) to compare $\kappa_{gf}$ measured in this work with $\kappa_{ccn}$ measured by previous studies, and in this table the results reported by Gaston et al. (2017) were included.

2) We have rephrased our discussion on comparison between our and previous work, and further details can be found in Section 3.1.4 of the revised manuscript (line 482-497).

2. The authors are encouraged to create another table or a $\kappa$ plot with their $\kappa$(GF) values and to report these values in the abstract. This will attract more attention to their work.

**Reply:** The following changes have been made in the revised manuscript, as suggested:

1) A table (Table 5, Page 26) has been created to compare $\kappa_{gf}$ measured in this work with $\kappa_{ccn}$ measured by previous studies;

2) We have also mentioned $\kappa_{gf}$ values in the abstract (line 40-43).

Conclusions:

1. The authors are encouraged to point out the broader implications of their work for the water uptake and cloud nucleating properties of fresh and processed dusts and sea salts.

**Reply:** This is a very good point. In the revised manuscript we have changed the title of Section 4 to "Summary and Conclusion", since this section is more like a summary of our work.

Furthermore, we have added a new section (Section 3.3.2, line 666-700), entitled "Atmospheric implication", to discuss atmospheric implications of our work for mineral dust, saline mineral dust and sea salt aerosol. Please refer to Section 3.3 in the revised manuscript for details.

2. The authors are encouraged to also point out the ability of the Clausius-Clapeyron equation to predict the temperature-dependent behavior of the water uptake properties of some of the salts.

**Reply:** As suggested, the following change has been made in the revised manuscript (line 708-709): "…both showing negative dependence on temperature, and the dependence of their DRH on temperature can be approximated by the Clausius-Clapeyron equation."

---

## Author Comment (AC4) · 23 Sep 2018

Comments by Referees are in blue.

Our replies are in black.

Changes to the manuscript are highlighted in red both in here and in the revised manuscript.

**Reply to Ref #1**

**General comments:**

The authors presented a comprehensive study of the hygroscopic properties of calcium- and magnesium- containing salts using a vapor sorption analyzer and a HTDMA. The change of sample mass with RH and the corresponding DRH value was reported for these eight compounds together with their hygroscopic growth factor values at 90% RH. The dataset is rich, however, the comparison and discussion is not sound that major revisions are needed. The manuscript may be acceptable for publication in ACP after the following concerns are fully addressed.

**Reply:** We would like to thank Ref #1 for his/her insightful and detailed comments, which have largely helped us improve our manuscript. We have addressed all the comments adequately in the revised manuscript, as detailed below.

**Major comments:**

1. This work used a vapor sorption analyzer to measure the change of sample mass with relative humidity and the deliquescence relative humidity of eight different compounds. However, as I understand, the materials the author used in this study are not atmospheric particles, but actually bulk samples. Please clarify how these results represent atmospheric conditions. The hygroscopic properties and DRH of aerosol particles would probably deviate significantly from that of bulk samples. Please extend your manuscript with explicit discussions regarding these issues to prove the significance for atmospheric research.

**Reply:** In addition to this comment (major comment #1), similar concerns have been raised in a few other comments (including major comment #2, part of major comment #3, major comment #4, and specific comment #12) regarding the atmospheric relevance of our VSA results and the comparison between the VSA and H-TDMA measurement. In response to these comments, we have added one section (Section 3.3.1, line 635-665), entitled "Comparison between H-TDMA and VSA measurements". Here we response to these comments and also outline changes implemented in the revised manuscript.

1) Indeed bulk samples, instead of aerosol particles, were used in our VSA experiments. However, thermodynamic principles which govern the equilibrium between water vapor and liquid water in the aqueous solutions are the same for bulk samples and atmospheric particles (we note that particle size may play a role for aerosol particles due to the Kelvin effect, while bulk samples have flat surfaces). More specifically, when RH are higher than DRH, bulk samples (as well as aerosol particles) would be deliquesced to form aqueous solutions, and the water-to-solute ratios would depend on RH. Water-to-solute ratios measured in our work for aqueous solutions formed from the eight compounds, can be used to validate aerosol thermodynamic models which are widely employed to predict aerosol hygroscopicity based on aerosol chemical composition, measured or prescribed.

2) For RH above the DRH, both bulk samples and aerosol particles would become deliquesced to form aqueous solutions; at a given RH, the two types of solutions would have the same concentrations. VSA measured the mass of aqueous solutions while H-TDMA measured the diameter (which could be used to calculate the volume), and therefore the two types of results can be related to each other via solution densities, which also depend on RH. In addition, diameter changes measured by H-TDMA can also be used to validate aerosol thermodynamic models, which can be used to calculate the RH-dependent volume of aerosol particles.

3) When RH is below the DRH, discrepancies are found for the two types of measurements. This is because bulk samples and aerosol particles are of different crystalline state under dry conditions, as we discussed in our original manuscript. For this RH range, H-TDMA results should be used for atmospheric implications since aerosol particles used in H-TDMA measurements are of direct atmospheric relevance. However, the states of aerosol particles at RH below DRH (i.e. before full deliquescence) remain largely unknown. In the revised manuscript (line 656-665) we have outlined relevant remaining open questions and discussed how they can be addressed in future work.

2. What is the relation between the mass growth factor and mobility growth factor as measured by two independent methods? Extensive works have been performed to measure the hygroscopic growth factor of atmospheric relevant compounds from previous studies. How to compare the mass growth factor obtained by the vapor sorption analyzer with their results and what kind of uncertainties should be taken into account?

**Reply:** In our reply to major comment #1, we have also addressed major comment #2 together. We would kindly refer the reviewer to our reply to major comment #1 for further information.

3. The DRH values of these studied compounds can also be measured by your HTDMA setup. Why the results obtained by your HTDMA did not agree with the ones from the VSA. What is the explanation for the discrepancies? Since the VSA measures the bulk samples, are these results obtained from the VSP measurements applicable in atmospheric research? Could you also please plot the GF-PDF for each compounds measured by the HTDMA? Are there unimode or bimode for your growth factor distributions at different RHs?

**Reply:** 1) The difference in these two types of experiments can be explained by the fact that samples used in the two types of experiments were of different state: samples used in VSA measurement were crystalline, while aerosol particles used in H-TDMA experiments were amorphous. Explanations and discussions have been provided in our original manuscript (line 342-350, page 19, Section 3.1.4; line 488-499, page 26-27, Section 3.2.2); in addition, we have also added a few sentences (line 643-647) in the revised manuscript to further explain such difference and to underscore such differences.

2) Since major comments #1 also raised similar concerns on the atmospheric relevance of our VSA measurements, please refer to our reply to major comment #1 for further details of our discussion on the atmospheric relevance of our VSA results.

3) The size distribution of aerosol particles at different RH are unimode. In the supplementary information of revised manuscript we have provided the size distribution of $Ca(NO_3)_2$ at three RH as an example to illustrate it, since we feel it is not necessary to plot the size distributions for all the eight types of aerosol we studied. In addition, in the revised manuscript we have added one sentence (line 146-149) to summarize the main point regarding size distributions: "Size distributions of all the eight types of aerosol particles, measured using the SMPS, were found to be unimode, as illustrated by Figure S1 (in the supplementary information) in which size distributions of $Ca(NO_3)_2$ aerosols at 4, 50 and 90% RH are displayed as an example."

4. For the inorganic species you studied, you stated they are important components in mineral dust or sea salt particles. However, for the VSA measurements, you studied their hygroscopic properties of their hydrate forms, while you measured the HGF of these compounds in their anhydrous state using HTDMA. As I see, the hygroscopic properties of these compounds vary significantly

between their anhydrous states and hydrate states (for instance, line 420-421). I feel difficult to relate your results with your introduction and objectives. Which state exist in the real atmosphere? Moreover, which state is hygroscopic and which values should we use for further study? Please clarify and be consistent through your whole manuscript. Otherwise, give your explanations.

**Reply:** In our reply to major comment #1, we have also addressed major comment #4 together. We would kindly refer the reviewer to our reply to major comment #1 for further information.

5. For your conclusion part, it is more like a summary of your results without any atmospheric implications. Please rephrase it and highlight its atmospheric applications.

**Reply:** We fully agree with the referee. In the revised manuscript we have changed the title of Section 4 to "Summary and Conclusion", since this section is more like a summary of our work; furthermore, we have added a new section (Section 3.3.2, line 666-700), entitled "Atmospheric implications", to discuss the atmospheric implications of our work. Please refer to Section 3.3.2 in the revised manuscript for details.

**Specific comments:**

1. Line 93-97: These two statements are in conflict with each other.

**Reply:** In the revised manuscript we have rephrased these two sentences (line 98-102) to make what we mean clearer: "However, only a few previous studies explored hygroscopic growth of $Mg(CH_3COO)_2$ and $Ca(CH_3COO)_2$, using techniques based on bulk samples (Wang et al., 2005; Ma et al., 2012; Pang et al., 2015). To our knowledge, hygroscopic growth factors have never been reported for $Ca(HCOO)_2$, $Mg(HCOO)_2$, $Ca(CH_3COO)_2$ and $Mg(CH_3COO)_2$ aerosol particles."

2. Line 128-131: How long is your humidifier and what is the flow rate? And what is the accuracy of your RH measurements, please give its uncertainty.

**Reply:** The flow rate was 300 mL/min, and the residence time in the humidification section was ~27 s, and the accuracy in RH measurement was ±0.8%. In the revised manuscript, we have provided flow rate information (line 127-128): "After exiting the atomizer, an aerosol flow (300 mL/min) was passed through a Nafion dryer…" as well as information on the humidifier and RH measurement (line 131-135): "…the aerosol flow was transferred through a humidification section with a residence time of ~27 s to be humidified to a given RH. The humidification section was made of two Nafion humidifiers (MD-700-12F-1, Perma Pure) connected in series. The RH of the

resulting aerosol flow was monitored using a dew-point meter, which had an absolute uncertainty of ±0.8% in RH measurement as stated by the manufacturer (Michell, UK). After humidification..."

3. Line 139: What do you mean by a particle sample? Did you generate particles and measure the mass of these particles? If not, please rephrase it.

**Reply:** Samples used in the VSA experiments were bulk samples, which can be powdered particles or small crystalline grains. In the revised manuscript (line 160) we have changed "a particle sample" to "a bulk sample" to be more accurate.

4. Line 210-211: What is the possible reason for the deviations?

**Reply:** Since all the other studies agree very well, and the results reported by Apelblat (1992) are always 3-5% higher. This may suggest that the study by Apelblat (1992) could have unknown systematical error. In the revised manuscript we have added one sentence (line 241-242) to explain the possible reason for this deviation: "This may imply that water vapor pressure measurements by Apelblat (1992) could have unknown systematical errors."

5. Line 222-223: Give proper reference for Eq. 1 in your manuscript, the original source but not only these who also cites it. In addition, which solubility (at which temperature condition) you used for your calculation, as it also depends on temperature.

**Reply:** 1) In the revised manuscript (line 255), we have cited two additional references (Wexler and Seinfeld, 1991; Seinfeld and Pandis, 2006) which detailed the process to derive this equation.

2) For this equation to be valid, the solubility and the enthalpy of dissolution are assumed to be constant for the temperature range considered. In the revised manuscript we have added one sentence (line 264-266) to clarify it: "It should be noted that for Eq. (2) to be valid, both the enthalpy of dissolution and the water solubility are assumed to be constant for the temperature range considered."

6. Line 225: Enthalpy of what? Deliquesce or dissolution?

**Reply:** It should be "enthalpy of dissolution". We have changed it in the revised manuscript (line 258).

7. Line 246: Could you give proper explanation why WSR increase with a decreasing in temperature.

**Reply:** This is because the dissolution processes are exothermic. In the revised manuscript (line 281-285), we have added one sentences to explain such dependence: "As discussed in Section

3.1.1, the enthalpies of dissolution ($\Delta H_s$) are negative for these compounds, suggesting that their dissolution processes in water are exothermic; therefore, dissolution is favored at lower temperatures and at a given RH, decrease in temperature would lead to increase in WSR in the aqueous solutions."

8. Line 253, Table 2: How could you get the WSR value for $Ca(NO_3)_2$ at 50% RH, as it did not deliquesce yet according to your previous results in Table 1 (DRH as 60.5%).

**Reply:** The referee is right. We made an error when we calculated WSR for $Ca(NO_3)_2$ at 5 °C, and in fact at 5 °C we could calculate WSR for RH at or above 70%. In the revised manuscript we have corrected this error.

9. Line 258-261: I don't understand the sentences.

**Reply:** In the revised manuscript (line 297-299), we have expanded these sentences to provide further explanation and to increase the readability: "Water activities of $Ca(NO_3)_2$ solutions at 25 °C were measured to be 0.904, 0.812 and 0.712 when the concentrations were 2.0, 3.5, and 5.0 mol kg$^{-1}$, respectively (El Guendouzi and Marouani, 2003). Since water activity of a solution is equal to the RH of air in equilibrium with the solution, it can be derived that the molality concentrations of $Ca(NO_3)_2$ solution were 2.0, 3.5, and 5.0 mol kg$^{-1}$ when RH was 71.2, 81.2 and 90.4%; in other words, WSR were found to be 11.1, 15.9 and 27.8 at 71.2, 81.2 and 90.4 % RH, respectively (El Guendouzi and Marouani, 2003)."

10. Line 265: What concentration do you mean here? Bulk solution or droplet? Please be specific.

**Reply:** To be more specific, in the revised manuscript (line 305) we have changed "…the concentrations…" to "…the concentrations of the bulk solutions…".

11. Line 286: Did you also observe similar phenomena for the other two inorganic compounds for phase transition, as it seems to be according to Fig. 1 in your manuscript.

**Reply:** For compounds investigated in our work, solid-solid phase transition was only observed for $CaCl_2$.

12. Line 311-312: I don't think this is fairly new result as it is still bulk sample. We should always consider size effect as it is atmospheric or at least particle-relevant.

**Reply:** Regarding to our response to this comment, please refer to our reply to major comment #1 for further information.

13. Line 337: What is the stuff after atomizing? Are they in hydrate state or not?

**Reply:** These aerosol particles, generating by atomizing water solutions, are likely to be amorphous. This was explained in line 343-350 in the original manuscript.

14. Line 363-365 and line 379-382: So should we use the dry diameter selected by the DMA (100 nm) or not. If yes, it seems your results did not agree with the ones from Gibson et al., (2006) in line 363-365, but agreed in line 379-382. Please clarify.

**Reply:** We believe the dry particle diameter (100 nm) should be used to calculate GF measured by Gibson et al. (2006); and if used, our results show relative good agreement with them. In the revised manuscript we have rephrased relevant sentences (line 401-409) to make it clearer: "If the dry diameter selected using the DMA (i.e. 100 nm) was used for calculation instead, GF reported by Gibson et al. (2006) would be ~1.34 at 80% RH and ~1.58 at 85% RH; compared with our results (1.51±0.02 at 80% RH and 1.62±0.01 at 85% RH), GF reported by Gibson et al. (2006) are ~11% smaller at 80% RH and only ~3% smaller at 85%. In the second study (Jing et al., 2018), GF were determined to be 1.56 at 80% RH and 1.89 at 90% RH; compared with our results (1.51±0.02 at 80% RH and 1.79±0.03 at 90% RH), GF reported by Jing et al. (2018) were ~3% larger at 80% RH and ~6% larger at 90% RH. Overall, our results show reasonably good agreement with the two previous studies (Gibson et al., 2006; Jing et al., 2018)."

15. Extra cautions must be taken by introducing several scientific terms in the manuscript. For instance, in line 221-223. What is the scientific reason to study the temperature-dependence of DRH and its enthalpy value? Please clarify. For instance, in line 258-263, why you studied the water-solute ratio and what is this variable used for? What is the relation between water activity and water-solute ratio? And how you converted it to each other in details?

**Reply:** In the revised manuscript we have added one sentence (line 252-253) to clarify the scientific reason to study temperature dependence of DRH values: "Temperature in the troposphere varies from ~200 to >300 K, and it is thus warranted to explore the effects of temperature on hygroscopic properties of atmospherically relevant particles."

The water activity of a solution with flat surface is equal to the RH of the air in equilibrium with the solution; therefore, if we know the water to solute ratios as a function of water activity, we can calculate water to solute ratios of aerosol particles and aerosol water content as a function of RH. We have added one new section (Section 3.3, line 634-700) in the revised manuscript, entitled "Discussion and atmospheric implication", to further discuss our experimental results and

their atmospheric implications. Please refer to Section 3.3 in the revised manuscript for more details.

16. Which particle size did you selected during the HTDMA measurements? In Eq. 4 in your manuscript, where is the Kelvin term? Please use the correct formulation and make further comparison. For instance, in line 410-411, which particle size or which supersaturation they selected in their CCN measurements?

**Reply:** The mobility diameter of dry aerosol particles selected using the first DMA was always 100 nm, and in Section 2.2 of the revised manuscript, we have added one sentence (line 145-146) to clarify it.

The Kelvin effect is negligible for hygroscopic growth of aerosol particles with a dry diameter of 100 nm. In the revised manuscript (line 443-454) we have provided the original equation which takes into account the Kelvin effect and explained how Eq. (4) is derived. Please refer to the revised manuscript for details.

The dry particles sizes were in the range of 50-125 nm, and in the revised manuscript we have added one sentence (line 471-472) to provide relevant information: "In previous work which measured CCN activities measured (Sullivan et al., 2009; Tang et al., 2015; Gaston et al., 2017), the dry particle diameters used were typically in the range of 50-125 nm."

17. In addition, please rephrase your discussion part and make sound comparisons with the other studies. For instance, line 367-368, GF of $Ca(NO_3)_2$ aerosols was measured to be 1.79 in your work, while Jing et al., (2018) reported it to be 1.89 at 90%RH. In line 387-389, Park et al., (2009) measured the GF of $CaCl_2$ to be 1.59 at 90% RH and the measured value from your result was 1.71. There were some differences (around 7%) but not always in good agreement as you stated in the manuscript between your results and the ones from others. Please give proper discussions.

**Reply:** As suggested, in the revised manuscript we have rephrased our discussion when comparing our results with previous work, to make our statement more precise. For example, when comparing our measurement with the two previous studies for hygroscopic growth of $Ca(NO_3)_2$ aerosol particles, the discussion (line 401-409) has been changed to: "If the dry diameter selected using the DMA (i.e. 100 nm) was used for calculation instead, GF reported by Gibson et al. (2006) would be ~1.34 at 80% RH and ~1.58 at 85% RH; compared with our results (1.51±0.02 at 80% RH and 1.62±0.01 at 85% RH), GF reported by Gibson et al. (2006) are ~11% smaller at 80% RH and only ~3% smaller at 85%. In the second study (Jing et al., 2018), GF were determined to be

1.56 at 80% RH and 1.89 at 90% RH; compared with our results (1.51±0.02 at 80% RH and 1.79±0.03 at 90% RH), GF reported by Jing et al. (2018) were ~3% larger at 80% RH and ~6% larger at 90% RH. Overall, our results show reasonably good agreement with the two previous studies (Gibson et al., 2006; Jing et al., 2018)."

---

## Referee Report (RR1)

**General comments:**

Based on the HTDMA and VSA measurements, this work has reported a large dataset of hygroscopicity of Ca- and Mg-containing samples, and discussed the temperature-dependence of DRH values. The calculated $\kappa$ results were comparable to those of previous hygroscopic studies. This can enrich the knowledge of water uptake by Ca- and Mg-containing mineral dust and sea salts in the troposphere. The quality of this manuscript could be largely improved, providing a better organization of the main contents with clearer and more concise descriptions. A native or professional speaker is recommended for the language revision including a thorough grammar check. The presented work has its scientific importance, while major revision is suggested before the final publication.

**Specific comments:**

1. Page 8, line 175: The authors have suggested a "*baseline drift*" for the determination of the mass growth factor, or rather the DRH value. However, it is confusing to readers that how "*a significant increase in sample mass*" was defined. What is the corresponding reference value for the investigated salts? Does the baseline drift in sample mass always correspond to a same value for different types of Ca- and Mg-containing samples?

2. Page 8, line 176: Experiments are repeatable, but how "each measurement" could be repeated?

3. Page 8, line 180: It is unclear that when did "*an equilibrium*" or "*a new equilibrium*" state actually reach during the experiments, at least it is difficult to tell from the corresponding figures, e.g., it is still able to expect some increase in the sample mass as displayed in Fig.5.

   In addition, the determination of an equilibrium condition will directly influence the accuracy of water-to-solute ratio, mass growth factor, and DRH results reported in this work. In this sense, how to quantify the influence of this "equilibrium" state on measured change in $m/m_0$ or in normalized sample mass with varying RH, and further on DRH and comparison of different hygroscopic results?

4. Page 9, line 188: Following the comment #3, there is no definition of "normalized sample mass", which is also important for the determination of DRH.

   I feel lost in the **authors' response to Specific comment #15 of Ref #3**, which explained that "*When plotting these three figures (i.e., Figs.1, 3, and 5), we did not*

*always to normalize the sample mass to the dry particle mass. This is why the right y-axis in these three figures is labelled as normalized sample mass instead of m/m₀.*" How to quantify the change in sample mass accurately, if the "normalized sample mass" was not always normalized by the initial dry mass? Accordingly, which kinds of mass were used for the normalization if "not always" to normalize, and how should the readers evaluate the comparability among these normalized sample mass results with increasing RH if without a constant comparison standard/baseline in the same experiment?

5.  Page 9, line 193: The authors have stated that "*such a large mass increase cannot be solely caused by water adsorption*". Why? Can you provide some clues or evidence to clarify this point, instead of making an assertion without any proof?

6.  Page 12, line 227: Is the corresponding RH determined to be 57±5%? Accordingly, can you really tell that they are "*in broad consistence with*" the results reported in this work?

7.  Page 12, line 242: In the expression of "*could have unknown systematical errors*", I was confused by the "unknown" here, as this sentence was supposed to provide the specific reasons. Besides, why can you assume that it is due to systematical errors if it's even unknown?

8.  Page 14, line 275

    "*Small changes in m/m₀ (typically <2%) were observed for some compounds (as shown in Tables 2 and 6) when RH was below corresponding DRH values, mainly due to water adsorption/desorption and baseline drift.*"

    This sentence sounds somewhat ambiguous. What does the "small changes" refer to, the increase or decrease in mass change? How to relate the small mass change with the corresponding causes, i.e., water adsorption, water desorption, and baseline drift?

9.  Page 16, line 296

    "*Water activities of $Ca(NO_3)_2$ solutions at 25 °C were measured to be 0.904, 0.812 and 0.712 when the concentrations were 2.0, 3.5 and 5.0 mol $kg^{-1}$, respectively*".

    The results might be wrongly sequenced in either RH values or molality concentrations, as which are opposite to the corresponding details included in the following sentence. To avoid such mistakes, please also check for the $Mg(NO_3)_2$ sample and data results elsewhere carefully.

10. Page 16, line 297: Please confirm the physical meaning of "water activity" and "RH of air".

11. Page 18, line 330: How was the mass increase (i.e., 48±7%) determined here, which seemed not reflected in Fig.3? What is the baseline drift or reference mass value used for the normalization of sample mass?

12. Page 24, line 430: Try to rephrase the whole sentence starting with "For comparison" into a more concise one, so does the beginning sentence in the following paragraph. This will help the readers to catch the point more easily.

13. Page 24, line 443: In the section of "Comparison between hygroscopic growth with CCN activities", equations used for the $\kappa_{gf}$ calculation and relevant details belong to theoretical methodologies other than observed results. It would make more sense to reorganize the corresponding contents, e.g., into the Experimental section.

14. Table 5: Corresponding references are needed for "*previous studies*".

15. Page 28, line 519: How much does the "*significant increase*" refer to? Line 521: Why is the DRH determined to be 90.5±1.0% instead of ±0.5%? A similar issue existed in Line 533: "giving a measured DRH of 71.5±1.0%". Please also have a check on the corresponding values such as in the Conclusion section.

16. Page 32, line 585: The last sentence is really confusing. What is the connection between "*GF is typically expected to increase with RH*" and "*small decrease in GF (~0.03) for RH between 50 and 70% may reflect uncertainties in GF measurements*"? What kinds of uncertainties in GF measurements are you trying to suggest? What is the possible influence of particle phase state on measured growth factors?

    Besides, what does it mean that "*HTDMA measurements cannot resolve a GF difference as small as 0.03*", since the presented GF results (i.e., 1.03±0.01 and 1.00±0.01) were actually measured with HTDMA and the uncertainties in the corresponding GF values are as small as ±0.01? A more scientific/convincing explanation for this observed decrease in GF will need to be provided.

17. Page 34, line 635: This Sect.3.3.1 was designed to compare the hygroscopicity results between HTDMA and VSA measurements. However, the organization of the whole section was a bit misleading. It seems that only one sentence is directly related to the topic (i.e., "To directly link the mass change (measured using VSA) with diameter change (measured using H-TDMA), solution densities, which also vary with RH, are needed."). I didn't see more details about how these two types of growth factors were actually reconciled.

    Although the authors have tried to explore the atmospheric relevance of their experimental results (e.g., applicability to the conditions when RH is below or exceeds DRH), it's important to distinguish the difference in hygroscopicity for both bulk powder samples and individual particles. For clearer clarification, such information might be necessary before detailing the corresponding water uptake results.

Line 662-663: This sentence is confusing. Whether is it suggesting a possible way for the future investigations of the particle phase state with changing RH, or is it trying to answer the open question with the results obtained from previous studies? Then what is the conclusion or possible answers to the question according to previous work?

A similar issue also exists in the last two sentences of this section. Haven't you already measured the water-to-solute ratios and mass growth factors in this study? Why not trying to address the question using your own experimental data?

---

## Author Response (AR2)

Comments by Referees are in blue.
Our replies are in black.
Changes to the manuscript are highlighted in red both here and in the revised manuscript.

**General comments:**
Based on the HTDMA and VSA measurements, this work has reported a large dataset of hygroscopicity of Ca- and Mg-containing samples, and discussed the temperature-dependence of DRH values. The calculated $\kappa$ results were comparable to those of previous hygroscopic studies. This can enrich the knowledge of water uptake by Ca-and Mg-containing mineral dust and sea salts in the troposphere. The quality of this manuscript could be largely improved, providing a better organization of the main contents with clearer and more concise descriptions. A native or professional speaker is recommended for the language revision including a thorough grammar check. The presented work has its scientific importance, while major revision is suggested before the final publication.

**Reply:** We would like to thank ref #3 for reviewing our manuscript. His/her comments have been carefully addressed in our revised manuscript, as detailed below. In addition, we have asked a professional editor to edit our manuscript thoroughly.

**Specific comments:**
1. Page 8, line 175: The authors have suggested a "baseline drift" for the determination of the mass growth factor, or rather the DRH value. However, it is confusing to readers that how "a significant increase in sample mass" was defined. What is the corresponding reference value for the investigated salts? Does the baseline drift in sample mass always correspond to a same value for different types of Ca- and Mg- containing samples?

**Reply:** Here we state that the DRH was RH at which a significant increase in sample mass occurred. The mass measurement was affected by signal noise and baseline drift; only when the mass increase was significant when compared to signal noise and baseline drift, the mass increase was due to the occurrence in deliquescence. As shown in Figure 1a for example, when RH was increased from 52 to 53%, the mass was increased by 15% and was still increasing; for comparison, the baseline drift was <2%. Therefore, we concluded that deliquescence occurred at 52-53% RH.

2. Page 8, line 176: Experiments are repeatable, but how "each measurement" could be repeated?

**Reply:** As suggested, in the revised manuscript (page 8, line 176) we have changed "measurement" to "experiment".

3. Page 8, line 180: It is unclear that when did "an equilibrium" or "a new equilibrium" state actually reach during the experiments, at least it is difficult to tell from the corresponding figures, e.g., it is still able to expect some increase in the sample mass as displayed in Fig. 5. In addition, the determination of an equilibrium condition will directly influence the accuracy of water-to-solute ratio, mass growth factor, and DRH results reported in this work. In this sense, how to quantify the influence of this "equilibrium" state on measured change in $m/m_0$ or in normalized sample mass with varying RH, and further on DRH and comparison of different hygroscopic results?

**Reply:** As we stated in the original manuscript (page 8, line 177-179), an equilibrium was reached when the mass change was <0.1% in 30 min; nevertheless, the time required to reach an equilibrium varied in different experiments, depending on RH, sample mass, and compounds under investigation. Indeed the mass was still increasing in Figure 5a. However, the total mass change was very small (by ~3%) because the hygroscopicity of $Ca(HCOO)_2$ was very low, and further increase in mass, given enough time, is expected to be very small.

Since an equilibrium was only reached when the mass change was <0.1% in 30 min, we expect that $m/m_0$ has an uncertainty of ~0.1% or even smaller, and such uncertainty was negligible for deliquesced samples. WSR values measured in this and our previous work (Gu et al., 2017a; Gu et al., 2017b; Jia et al., 2018) agree well with those reported by others for a number of compounds, further suggesting that our criterion used to judge whether an equilibrium was reached (as well as our experimental method) is robust.

4. Page 9, line 188: Following the comment #3, there is no definition of "normalized sample mass", which is also important for the determination of DRH. I feel lost in the authors' response to Specific comment #15 of Ref #3, which explained that " When plotting these three figures (i.e., Figs.1, 3, and 5), we did not always to normalize the sample mass to the dry particle mass. This is why the right y -axis in these three figures is labelled as normalized sample mass instead of $m/m_0$." How to quantify the change in sample mass accurately, if the "normalized sample mass" was not always normalized by the initial dry mass? Accordingly, which kinds of mass were used for the normalization if "not always" to normalize, and how should the readers evaluate the comparability among these normalized sample mass results with increasing RH if without a constant comparison standard/baseline in the same experiment?

**Reply:** When we plot the figures, we do not always normalize the sample mass to the dry particle mass; in fact, when we show the change of RH and sample mass with time, we can even present the actual sample mass without any normalization.

We agree with the referee that it is vital to normalize the sample mass to its dry mass if we want to quantify the change in sample mass due to water uptake. It is why we always normalized the sample mass to its dry mass (i.e. $m/m_0$), as we explained in the original manuscript (Tables 2, 3 and 6), when we calculate the change in sample mass.

5. Page 9, line 193: The authors have stated that "such a large mass increase cannot be solely caused by water adsorption". Why? Can you provide some clues or evidence to clarify this point, instead of making an assertion without any proof?

**Reply:** In the revised manuscript (page 9, line 193-194) we have added one sentence to clarify our claim: "such a large mass increase cannot be solely caused by water adsorption, since the mass of several monolayers of adsorbed water is estimated to be <1% of the dry particle mass (Gu et al., 2017b)."

6. Page 12, line 227: Is the corresponding RH determined to be 57±5%? Accordingly, can you really tell that they are "in broad consistence with" the results reported in this work?

**Reply:** Yes, the DRH was determined to be 57±5%, and it is more accurate to state that this value is slightly larger than that reported in our work. The following change was made in our revised manuscript (page 12, line 228-230): "In another study (Al-Abadleh et al., 2003), RH over the saturated $Ca(NO_3)_2 \cdot 4H_2O$ solution was measured to be 57±5% at room temperature; in other words, Al-Abadleh et al. (2003) reported a DRH of 57±5% for $Ca(NO_3)_2 \cdot 4H_2O$, slightly larger than that (49.5±1.0% at 25 oC) determined in our work."

7. Page 12, line 242: In the expression of "could have unknown systematical errors", I was confused by the "unknown" here, as this sentence was supposed to provide the specific reasons. Besides, why can you assume that it is due to systematical errors if it's even unknown?

**Reply:** Since the values reported by Apelblat (1992) are always larger than those reported by other studies, we suspect that the measurements by Apelblat (1992) may have some systematical errors; nevertheless, the exact reason is not clear. Therefore, we stated in the original manuscript "This may imply that water vapor pressure measurements by Apelblat (1992) could have unknown systematical errors."

8. Page 14, line 275: "Small changes in m/m0 (typically <2%) were observed for some compounds (as shown in Tables 2 and 6) when RH was below corresponding DRH values, mainly due to water adsorption/desorption and baseline drift." This sentence sounds somewhat ambiguous. What does the "small changes" refer to, the increase or decrease in mass change? How to relate the small mass change with the corresponding causes, i.e., water adsorption, water desorption, and baseline drift?

**Reply:** To response to this comments, in the revised manuscript (page 14, line 277) we have changed "small changes" to "small increases" to make it more precise. The small mass increase (typically <2%) occurred before deliquescence, very likely due to water adsorption and baseline drifts.

9. Page 16, line 296: "Water activities of $Ca(NO_3)_2$ solutions at 25 °C were measured to be 0.904, 0.812 and 0.712 when the concentrations were 2.0, 3.5 and 5.0 mol kg$^{-1}$, respectively." The results might be wrongly sequenced in either RH values or molality concentrations, as which are opposite to the corresponding details included in the following sentence. To avoid such mistakes, please also check for the $Mg(NO_3)_2$ sample and data results elsewhere carefully.

**Reply:** The RH values were incorrectly sequenced in the original manuscript, and it has been corrected in our revised manuscript (page 16, line 301). We have also carefully checked data elsewhere.

10. Page 16, line 297: Please confirm the physical meaning of "water activity" and "RH of air".

**Reply:** Water activity and RH are both basic parameters in aerosol hygroscopicity, and we feel these two terms do not need further explanation for readers interested in our manuscript.

11. Page 18, line 330: How was the mass increase (i.e., 48±7%) determined here, which seemed not reflected in Fig.3? What is the baseline drift or reference mass value used for the normalization of sample mass?

**Reply:** As shown in Figure 3, when RH was increased from 0% to 20%, the normalized sample mass was increased from 1 to ~1.5; therefore, the mass was increased by ~50% (48±7%, more exactly).

12. Page 24, line 430: Try to rephrase the whole sentence starting with "For comparison" into a more concise one, so does the beginning sentence in the following paragraph. This will help the readers to catch the point more easily.

**Reply:** We believe that it is necessary to provide exact values reported by the two studies in the comparison so that the readers would know the details. Nevertheless, as suggested by the referee, in the revised manuscript (page 24, line 432) we have rephrased these two sentences to further increase the readability.

13. Page 24, line 443: In the section of "Comparison between hygroscopic growth with CCN activities", equations used for the $\kappa_{gf}$ calculation and relevant details belong to theoretical methodologies other than observed results. It would make more sense to reorganize the corresponding contents, e.g., into the Experimental section.

**Reply:** We agree that some of this section can be moved into the experimental section. Nevertheless, this section is the first place where we introduce $\kappa$ values and in this section we also discuss the assumptions/caveats used in $\kappa_{gf}$ calculation; therefore, we believe that it is also suitable to leave details related to $\kappa_{gf}$ calculation in this section.

14. Table 5: Corresponding references are needed for "previous studies".

**Reply:** In this table when we use $\kappa$ values reported by previous studies, we cite the corresponding references. Therefore, it is not necessary to cite these references in the table caption.

15. Page 28, line 519: How much does the "significant increase" refer to?

**Reply:** To make it more clearer, in the revised manuscript (page 28, line 523) we have expanded the sentence to "In further experiments significant increase in sample mass (by >10%, and the sample was still increasing sharply when the experiment was terminated) was observed when RH was increased from 90 to 91% for Ca(CH$_3$COO)$_2$·H$_2$O at 25 $^o$C,"

Line 521: Why is the DRH determined to be 90.5±1.0% instead of ±0.5%? A similar issue existed in Line 533: "giving a measured DRH of 71.5±1.0%". Please also have a check on the corresponding values such as in the Conclusion section.

**Reply:** Our RH measurements have an uncertainty of ±1.0%, as stated in Section 2.2; therefore, we believe that the uncertainty of our measured DRH values should be ±1.0% or larger.

16. Page 32, line 585: The last sentence is really confusing. What is the connection between "GF is typically expected to increase with RH" and "small decrease in GF (~0.03) for RH between 50 and 70% may reflect uncertainties in GF measurements"? What kinds of uncertainties in GF measurements are you trying to suggest? What is the possible influence of particle phase state on measured growth factors? Besides, what does it mean that "HTDMA measurements cannot resolve a GF difference as small as 0.03", since the presented GF results (i.e., 1.03±0.01 and 1.00±0.01) were actually measured with HTDMA and the uncertainties in the corresponding GF values are as small as ±0.01? A more scientific/convincing explanation for this observed decrease in GF will need to be provided.

**Reply:** We agree with the referee that the decrease in GF may be related to particle phase state, and more specifically, caused by change in particle morphology (i.e. particle restruturing); in addition, the small change in GF (~0.03) may not be significant compared to the uncertainties in H-TDMA measurements, and the uncertianty (~0.01) in our reported GF is only the statistical error for three repeated experiments. In responce to this comment, we have changed this sentence in the revised manuscript (page 32, line 589-592) to "The decrease in GF may be caused by restructuring of particles or change in particle morphology (Vlasenko et al., 2005;Koehler et al., 2009); in addition, the small change in GF (~0.03) may not be significant when compared to the uncertianties in our H-TDMA measurements."

17. Page 34, line 635: This Sect.3.3.1 was designed to compare the hygroscopicity results between HTDMA and VSA measurements. However, the organization of the whole section was a bit misleading. It seems that only one sentence is directly related to the topic (i.e., "To directly link the mass change (measured using VSA) with diameter change (measured using H-TDMA), solution densities, which also vary with RH, are needed."). I didn't see more details about how these two types of growth factors were actually reconciled. Although the authors have tried to explore the atmospheric relevance of their experimental results (e.g., applicability to the conditions when RH is below or exceeds DRH), it's important to distinguish the difference in hygroscopicity for both bulk powder samples and individual particles. For clearer clarification, such information might be necessary before detailing the corresponding water uptake results.

**Reply:** We feel this section is well-organized, as described below: 1) in the first paragraph, we provide necessary background needed to compare two types of measurements; 2) in the second paragraph, we compare the two types of measurements for RH above the DRH; 3) in the third paragraph, we compare the two types of measurements for RH below the DRH.

As pointed out by the referee, it is important to distinguish the two types of samples, and in fact this aspect has been emphasized in our original manuscript (e.g., Section 3.3.1 and elsewhere). Line 662-663: This sentence is confusing. Whether is it suggesting a possible way for the future investigations of the particle phase state with changing RH, or is it trying to answer the open

question with the results obtained from previous studies? Then what is the conclusion or possible answers to the question according to previous work?

**Reply:** This sentence is to suggest a possible way for future investigation, and Li et al. (2017) presented a method which can be used to measure particle phase state. In the revised manuscript (page 36, line 668) we have rephrased this sentence to make it more clear: "In this aspect, measurements of particle phase state of $Ca(NO_3)_2$ and other aerosols considered in our work, using the appartus described previously (Li et al., 2017), can shed some light."

A similar issue also exists in the last two sentences of this section. Haven't you already measured the water-to-solute ratios and mass growth factors in this study? Why not trying to address the question using your own experimental data?

**Reply:** When RH is above the DRH of $Ca(NO_3)_2 \cdot 4H_2O$, $Ca(NO_3)_2$ aerosol particles are deliquesced and their WSR were measured in this work; however, when RH is below the DRH of $Ca(NO_3)_2 \cdot 4H_2O$, we do not know the WSR for $Ca(NO_3)_2$ aerosol particles. In the revised manuscript (page 36, line 669-670), we have expanded this sentence to make it more clearer: "Furthermore, how do water-to-solute ratios change with RH for $Ca(NO_3)_2$ aerosol particles when RH is below the DRH of $Ca(NO_3)_2 \cdot 4H_2O$?"

**References:**

Gu, W. J., Li, Y. J., Zhu, J. X., Jia, X. H., Lin, Q. H., Zhang, G. H., Ding, X., Song, W., Bi, X. H., Wang, X. M., and Tang, M. J.: Investigation of water adsorption and hygroscopicity of atmospherically relevant particles using a commercial vapor sorption analyzer, Atmos. Meas. Tech., 10, 3821-3832, 2017.

Gu, W. J., Li, Y. J., Tang, M. J., Jia, X. H., Ding, X., Bi, X. H., and Wang, X. M.: Water uptake and hygroscopicity of perchlorates and implications for the existence of liquid water in some hyperarid environments, RSC Adv., 7, 46866-46873, 2017.

Jia, X. H., Gu, W. J., Li, Y. J., Cheng, P., Tang, Y. J., Guo, L. Y., Wang, X. M., and Tang, M. J.: Phase transitions and hygroscopic growth of $Mg(ClO_4)_2$, $NaClO_4$, and $NaClO_4 \cdot H_2O$: implications for the stability of aqueous water in hyperarid environments on Mars and on Earth, ACS Earth Space Chem., 2, 159-167, 2018.

---

## Author Response (AR3)

January 2019

Atmospheric Chemistry and Physics

Dear Professor Hang Su,

Happy New Year! Thank you very much for handling our manuscript submitted to *Atmospheric Chemistry and Physics* (**MS No.:** acp-2018-412; **Title:** A comprehensive study of hygroscopic properties of calcium- and magnesium-containing salts: implication for hygroscopicity of mineral dust and sea salt aerosols).

We have addressed all the comments raised by you and the referee, and revised our manuscript very carefully and thoroughly. To proceed, we have uploaded three files, including 1) our reply to you and the referee; 2) the revised manuscript with changes highlighted in red; 3) the supplement to this manuscript.

On behalf of all the coauthors, I would like to thank you and referees for all the invaluable comments. Please feel free to contact me if you need any further information.

Sincerely,

Mingjin Tang, PhD

Guangzhou Institute of Geochemistry

Chinese Academy of Sciences

Email: mingjintang@gig.ac.cn

Comments by referees/editors are in blue.

Our replies are in black.

Changes to the manuscript are highlighted in red both here and in the revised manuscript.

1. Page 8, line 175: The authors have suggested a "baseline drift" for the determination of the mass growth factor, or rather the DRH value. However, it is confusing to readers that how "a significant increase in sample mass" was defined. What is the corresponding reference value for the investigated salts? Does the baseline drift in sample mass always correspond to a same value for different types of Ca- and Mg- containing samples? I think the referee was asking for the number or criteria, based on which you can conclude a "significant increase". Please specify it in a quantitative way, e.g. > signal noise + baseline drift, etc.

**Reply:** In our work the mass change due to signal noise and baseline drift was <0.5%, and in our DRH measurement when the sample was undergoing deliquescence we waited until the mass increase was >5% to ensure the occurrence of deliquescence. As suggested by the editor, in the revised manuscript we have included one sentence to provide additional information (==page 8, line 176-179==):"The measured relative change in sample mass due to signal noise and baseline drift was <0.5% in our work; in each experiment when we suspected that the sample were undergoing deliquescence at a certain RH, we did not stop the experiment until the mass increase was >5% to ensure the occurrence of deliquescence."

3. Page 8, line 180: It is unclear that when did "an equilibrium" or "a new equilibrium" state actually reach during the experiments, at least it is difficult to tell from the corresponding figures, e.g., it is still able to expect some increase in the sample mass as displayed in Fig. 5. In addition, the determination of an equilibrium condition will directly influence the accuracy of water-to-solute ratio, mass growth factor, and DRH results reported in this work. In this sense, how to quantify the influence of this "equilibrium" state on measured change in m/m0 or in normalized sample mass with varying RH, and further on DRH and comparison of different hygroscopic results? The referee is asking for the criteria to determine the equilibrium point, and how a non-equilibrium state could influence the accuracy of growth factor etc. This is in principle a general question for the proposed method and not specific for the investigated species, but hasn't been adequately addressed in the last round. What's the equilibrium time scale for your system and how did you estimate it? This is a very important issue, especially when viscous organics/semi-solid organics are involved or for measurements of new compounds, of which a reference gf is not available.

**Reply:** As pointed out by the referee and the editor, for hygroscopic growth studies (no matter techniques employed and sampled investigated), it is very important to ensure that the equilibrium is reached; otherwise the actual growth factors may be underestimated. In our experiments, as stated in the last paragraph of Section 2.2, if the sample mass change was <0.1% in 30 min at a given RH, the sampled was considered to reach the equilibrium.

However, if the sample mass was increasing very slowly and steadily, we may conclude erroneously that the system have reached the equilibrium even though the sample mass was still increasing. Therefore, we also need to inspect the data (sample mass as a function of time) to check whether the sample mass has reached the plateau. In the revised manuscript (page 8-9, line 181-185), we have added another two sentences to further explain the criterion we used to determine the equilibrium point: "At each RH the sample was considered to reach equilibrium with the environment when its mass change was <0.1% within 30 min, and RH was changed to the next value only after the sample mass was stabilized. If the sample mass was increasing steadily but with a very small rate (e.g., <0.1% in 30 min), the program we used may conclude erroneously that the system had reached the equilibrium; therefore, all the experimental data were inspected to check whether at each RH the sample mass reached the plateau (i.e. the system had reached the equilibrium)."

As noticed by the referee, though the mass change of $Ca(HCOO)_2$ was <0.1% in 30 min (as shown in Figure 5a in the original manuscript), the sample mass was actually still increasing, suggesting that the equilibrium was not reached yet. A similar issue was also identified for $Mg(HCOO)_2 \cdot 2H_2O$. We conducted additional experiments for these two compounds, and the mass increase at 95% RH was measured to be ~1.12 (instead of ~1.05 reported in our original manuscript) for $Ca(HCOO)_2$ and ~1.06 for $Mg(HCOO)_2 \cdot 2H_2O$. In the revised manuscript we have updated Figure 5a and relevant text (page 27-28, line 516-522; page 31, line 579-582; page 38, line 730-732; page 38, line 738-741) as well. Because Figure 5 has been updated, its caption has been changed to red in the revised manuscript. As shown in the updated Figure 5a, the sample mass was stabilized for 95% RH at the end of the experiment.

We have also checked experimental data collected using the vapor sorption analyzer for the other six compounds and confirmed that the equilibrium has been reached for these six compounds at every RH.

4. Page 9, line 188: Following the comment #3, there is no definition of "normalized sample mass", which is also important for the determination of DRH. I feel lost in the authors' response to Specific comment #15 of Ref #3, which explained that " When plotting these three figures (i.e., Figs.1, 3, and 5), we did not always to normalize the sample mass to the dry particle mass. This is why the right y-axis in these three figures is labelled as normalized sample mass instead of m/m0." How to quantify the change in sample mass accurately, if the "normalized sample mass" was not always normalized by the initial dry mass? Accordingly, which kinds of mass were used for the normalization if "not always" to normalize, and how should the readers evaluate the comparability among these normalized sample mass results? The referee has asked for a definition of "normalized sample mass", and how did you normalize your results if it is not normalized by the dry particle mass. These questions remain unanswered.

**Reply:** In previous versions of this manuscript, when we calculated water-to-solute ratios, we always normalized the sample mass to its dry mass. However, when we showed the experimental raw data (e.g., Figure 1a), we did not always normalized the sample mass to its dry mass; this did not change the actual results, but some of the figures would appear different (when compared to those in which sample mass was normalized to the dry mass) and be misleading to some extent.

To reduce confusion it may lead and increase the readability, as suggested by the referee and the editor, in the revised manuscript we have always normalized the sample mass to its dry mass, and relevant figures (Figures 1 and 3) have been updated. Since Figures 1 and 3 have been updated, their captions have been changed to red in the revised manuscript.

In addition, in the caption of Figure 1 we have included one sentence (page 10, line 211-212) to define the normalized sample mass: "In this paper the sample mass was always normalized to its dry mass."

7. Page 12, line 242: In the expression of "could have unknown systematical errors", I was confused by the "unknown" here, as this sentence was supposed to provide the specific reasons. Besides, why can you assume that it is due to systematical errors if it's even unknown? I agree with the referee and please rephrase the sentence.

**Reply:** Experimental details provided by Apelblat (1992) are so limited that it is difficult to know exactly why the reported DRH value are always slightly higher. We agree with the referee and the editor that it is arbitrary to state that the work by Apelblat (1992) could have systematical errors. In the revised manuscript (page 12, line 250-254) the sentence has been rephrased to "As shown in Table 1, DRH measured in our work agree very well with those reported by most of previous studies (Biggs et al., 1955; Robinson and Stokes, 1959; Al-Abadleh and Grassian, 2003; Rard et al., 2004), but are always 3-5% lower than those derived from Apelblat (1992). It is not clear why DRH values measured by Apelblat (1992) at different temperatures are always slightly higher than other studies."

Line 521: Why is the DRH determined to be 90.5±1.0% instead of ±0.5%? A similar issue existed in Line 533: "giving a measured DRH of 71.5±1.0%". Please also have a check on the corresponding values such as in the Conclusion section. I think the referee's concern came from Page 9 "Therefore, its DRH was measured to be 52.5±0.5 %".

**Reply:** Since RH in our VSA instrument had an absolute uncertainty of ±1.0%, the uncertainty in our measured DRH was assigned to be ±1.0%, instead of ±0.5%. In the revised manuscript (page 9, line 198-200), the sentence has been expanded for further clarification: "
[revised manuscript text omitted]

$46.0\pm1.0\%$ at 30 $^{\circ}$C for $Ca(NO_3)_2\cdot4H_2O$ and from $57.5\pm1.0\%$ at 5 $^{\circ}$C to $50.5\pm1.0\%$ at 30 $^{\circ}$C for

$Mg(NO_3)_2\cdot6H_2O$, both showing negative dependence on temperature, and this dependence can be approximated by the Clausius-Clapeyron equation. No significant dependence of DRH (around

31-33%) on temperature (5-30 $^{\circ}$C) was observed for $MgCl_2\cdot6H_2O$. $CaCl_2\cdot6H_2O$, found to deliquesce at ~28.5% RH at 25 $^{\circ}$C, exhibited complex phase transition processes in which

$CaCl_2\cdot2H_2O$, $CaCl_2\cdot6H_2O$ and aqueous $CaCl_2$ solutions were involved. Furthermore, DRH values were determined to be $90.5\pm1.0\%$ for $Ca(CH_3COO)_2\cdot H_2O$ and $71.5\pm1.0\%$ for

$Mg(CH_3COO)_2\cdot4H_2O$ at 25 $^{\circ}$C; for comparison, the sample mass was only increased by ~12% for

$Ca(HCOO)_2$ and ~6% for $Mg(HCOO)_2\cdot2H_2O$ when RH was increased from 0 to 95%, implying that the DRH of these two compounds were probably >95%.

We have also measured the change of sample mass as a function of RH up to 90% to derive the water-to-solute ratios (WSR) for deliquesced samples. WSR were determined at 25 and 5 $^{\circ}$C

for deliquesced $Ca(NO_3)_2\cdot4H_2O$, $Mg(NO_3)_2\cdot6H_2O$ and $MgCl_2\cdot6H_2O$ samples, and at 25 $^{\circ}$C for deliquesced $CaCl_2\cdot6H_2O$ and $Mg(CH_3COO)_2\cdot4H_2O$ samples. We found that compared to that at 0%

RH, large increases in sample mass only occurred when RH was increased from 90 to 95% for

$Ca(CH_3COO)_2\cdot H_2O$, and the WSR value was determined to be $5.849\pm0.064$ at 95% RH. Besides, deliquescence was not observed even when RH was increased to 95% for $Ca(HCOO)_2$ and

$Mg(HCOO)_2\cdot2H_2O$, and the ratios of sample mass at 95% to that at 0% RH, were determined to be for $1.119\pm0.036$ for $Ca(HCOO)_2$ and $1.064\pm0.020$ for $Mg(HCOO)_2\cdot2H_2O$. Despite that compounds investigated in the present work are important components for tropospheric aerosols, in general they have not been included in widely used aerosol thermodynamic models, such as E-

AIM (Clegg et al., 1998) and ISORROPIA II (Fountoukis and Nenes, 2007). The systematical and comprehensive datasets which we have obtained in this work are highly valuable and can be used to validate thermodynamic models if they are extended to include these compounds.

In addition, hygroscopic growth of aerosol particles was measured at room temperature for these eight compounds. Being different from solid samples for which the onset of deliquescence was evident, aerosol particles were found to grow in a continuous manner since very low RH (as low as 10%), implying that these dry aerosol particles generated from aqueous droplets were amorphous. Hygroscopic growth factors of aerosol particles at 90% RH were determined to be

$1.79\pm0.03$ and $1.67\pm0.03$ for $Ca(NO_3)_2$ and $Mg(NO_3)_2$, $1.71\pm0.03$ for both $CaCl_2$ and $MgCl_2$,

$1.54\pm0.02$ and $1.69\pm0.03$ for $Ca(HCOO)_2$ and $Mg(HCOO)_2$, and $1.26\pm0.04$ and $1.53\pm0.01$ 
[revised manuscript text omitted]

* Correspondence: Mingjin Tang (mingjintang@gig.ac.cn)

[Figure]

**Figure S1.** Number size distribution of Ca(NO$_3$)$_2$ aerosol particles (with a dry mobility diameter of 100 nm, as selected using a differential mobility analyzer) at <5, 50 and 90% RH.